# Direct observation of heterogeneous formation of amyloid spherulites in real-time by super-resolution microscopy

Min Zhang [1], Henrik D. Pinholt[1,2], Xin Zhou[3], Søren S.-R. Bohr [1], Luca Banetta[4], Alessio Zaccone [5], Vito Foderà [3✉] & Nikos S. Hatzakis [1,6✉]

Protein misfolding in the form of fibrils or spherulites is involved in a spectrum of pathological abnormalities. Our current understanding of protein aggregation mechanisms has primarily relied on the use of spectrometric methods to determine the average growth rates and diffraction-limited microscopes with low temporal resolution to observe the large-scale morphologies of intermediates. We developed a REal-time kinetics via binding and Photo-bleaching LOcalization Microscopy (REPLOM) super-resolution method to directly observe and quantify the existence and abundance of diverse aggregate morphologies of human insulin, below the diffraction limit and extract their heterogeneous growth kinetics. Our results revealed that even the growth of microscopically identical aggregates, e.g., amyloid spherulites, may follow distinct pathways. Specifically, spherulites do not exclusively grow isotropically but, surprisingly, may also grow anisotropically, following similar pathways as reported for minerals and polymers. Combining our technique with machine learning approaches, we associated growth rates to specific morphological transitions and provided energy barriers and the energy landscape at the level of single aggregate morphology. Our unifying framework for the detection and analysis of spherulite growth can be extended to other self-assembled systems characterized by a high degree of heterogeneity, disentangling the broad spectrum of diverse morphologies at the single-molecule level.

[1] Department of Chemistry, Faculty of Science, University of Copenhagen, Frederiksberg C 1871, Denmark. [2] Department of Physics and Institute for Medical Engineering and Science, Massachusetts Institute of Technology, Cambridge, MA 02139, USA. [3] Department of Pharmacy, Faculty of Health and Medical Sciences, University of Copenhagen, Copenhagen Ø 2100, Denmark. [4] Department of Applied Science and Technology, Polytechnic University of Turin, Torino, TO 10129, Italy. [5] Department of Physics, University of Milan, Milan 20122, Italy. [6] NovoNordisk Center for Protein Research, Faculty of Health and Medical Sciences University of Copenhagen, Copenhagen N 2200, Denmark. ✉email: vito.fodera@sund.ku.dk; hatzakis@chem.ku.dk

Protein misfolding is a hallmark of a number of devastating conditions, such as Alzheimer's and Parkinson's disease[1–3]. Moreover, insulin aggregates have been found deposited at diabetic patients' insulin injection sites[4]. As one of the important aggregates, amyloid spherulites have been found in the brain tissues in connection with the onset and progression of Alzheimer's disease[2,3]. In addition, they may also present opportunities to develop advanced materials for drug delivery[5]. Spherulites ranging from a few micrometers to several millimeters in diameter can form both in vivo and in vitro and are observed during the aggregation of multiple proteins including insulin, that we used as a model system here[3,6,7]. These aggregates are characterized by a fascinating core-shell morphology and seem to be the result of a general self-assembly process that is common to metal alloys[8], minerals[9], and polymers[10,11]. While we have a solid understanding of the fibrillar growth kinetics[12], the mechanisms of the formation and growth of spherulites are still limited[13–15].

The studies of protein spherulite formation primarily rely on spectrometric evidence for their average growth rates[13,16]. Considering that a high heterogeneity of aggregate populations may present within the same self-assembly reaction[17], bulk methods provide limited information on the aggregation kinetics of individual species, in the form of either fibrils or spherulites, averaging the effect of the morphological heterogeneity of the aggregate population. Fluorescent microscopy, as an intuitive and non-invasive method, has been used more and more in the area of protein aggregation to record the growth intermediates[15] and to observe the final structures[18]. While the direct observation of fibril growth and time lapses of spherulite growth with temporal resolution of minutes was recently reported for Aβ peptides[15,19–21], the kinetic analysis of individual aggregates mainly focuses on fibrils, which challenges the evaluation of the kinetics of the multiple and concurrent pathways. Meanwhile, due to the complex structures of spherulites, the resolution of diffraction-limited microscopy such as TIRFM or confocal microscopy is not sufficient to decipher the details of spherulites[15,22,23]. Super-resolution methods surpass the diffraction limitations, albeit often provide snapshots of the growth or the final morphology of fibrils[24–26] offering limited information on the temporal development of diverse structures.

Here we developed a method named Real-time kinetics via binding and Photobleaching LOcalization Microscopy (REPLOM) to directly observe the formation of individual protein amyloid structures using human insulin (HI) as a model system. The in vitro conditions leading to amyloid spherulites are identical to the ones used for the formation of fibrils[17,27]. Indeed the two species co-exist[17] and this is a further fact that justifies the need to have single-aggregate approaches to evaluate the heterogeneity of the system. Salt, pH, and protein concentration[28], the addition of alcohols[22], and surfactants[29] have been shown to contribute to the variability of the structure and morphology of spherulites. We choose the experimental conditions which guaranteed a time scale of the process allowing our real-time microscopy analysis. Attaining real-time videos of the spherulite growth process allowed us to reconstruct the super-resolution images of the spherulites and their growth kinetics. Using homemade software based on Euclidian minimum spanning tree and machine learning clustering[30–34], we quantitatively associated the growth rates to specific morphological transitions during growth, eventually extracting detailed energy barriers and, thus, the energy landscape for each type of aggregation morphology. Our data on astigmatism-based 3D direct stochastic optical reconstruction microscopy (dSTORM)[35], spinning disk confocal microscopy[36], and scanning electron microscopy (SEM) confirm that the presence of heterogeneous structures is not artifact of our method. Our combined results allowed us to differentiate among the different species in

solution and decipher the nature, morphology, and abundance of individual spherulites at different growth stages. Surprisingly, we found that HI spherulite growth is not exclusively isotropic and may occur anisotropically. According to their growth pathway and the final structures, we named them isotropic spherulites and anisotropic spherulites. We anticipate that the framework presented here will serve as a unique and generic methodology for the simultaneous detection and analysis of multiple species within a single self-assembly reaction. In the specific case of protein systems, the aggregation of which is related to degenerative diseases, our approach provides a platform for connecting kinetics, morphological transitions, and structure and further aids our understanding of interventions against degenerative diseases.

## Results and discussion

**Direct observation of diverse structures of HI spherulites by 3D dSTORM, SEM and spinning disk microscopy.** We thermally induced insulin amyloid aggregation using an established protocol[37] and examined the bulk kinetics by detecting the fluorescence of the amyloid-sensitive dye Thioflavin T (ThT) and the turbidity signal (Supplementary Fig. 1a). The kinetics traces at an incubation temperature of 60 °C, show the classical three-step profile, with the reaction reaching completion after only 3–4 h. The turbidity and ThT signals perfectly overlapped, suggesting that the aggregation reaction was entire of an amyloid-like origin[38]. Unlike previous studies[39,40], we found fluorophore labeling had no influence on the aggregation kinetics (Supplementary Fig. 1b). This may be because we used quite a low ratio of labeled insulin to unlabeled insulin monomer. Cross-polarized microscopy recordings of the characteristic Maltese cross, indicating spherulite formation under these conditions[7] (Supplementary Fig. 2). However, standard analysis of the bulk ThT signal was unable to provide information on the morphological transition occurring during the reaction.

To observe directly and with high-resolution the diverse structures of insulin aggregates, we combined the insights obtained from SEM and 3D dSTORM. Using 3D dSTORM allowed us to extend beyond diffraction-limited imaging by TIRF microscopy, which may mask spherulite structure details and growth directionality[35] (Fig. 1 and Supplementary Fig. 3). Direct comparison of 3D dSTORM recording with conventional read-outs using the same chromophore or ThT fluorescence, reveals that the distinct features of the aggregates are masked by diffraction limited imaging (Supplementary Fig. 3). Recordings at incubation times between 0.5 to 4 h provided direct recordings of the diverse early species that can co-exist at the same incubation time. We found spherical-like protein condensates of approximately 200 nm in diameter formed after 0.5 h, while a linear pattern was observed with incubation times ranging between 0.5 and 1 h. Surprisingly, the recordings beyond the diffraction limit revealed that at longer incubation times the commonly observed spherulites were found to co-exist in the mixture with anisotropically grown structures (Fig. 1 and Supplementary Figs. 4, S5c, and S5d).

The fact that both SEM and 3D dSTORM methods identified the same particle morphology supports this not to be an artifact of fluorescence microscopy, fluorophore labeling (Supplementary Fig. 1b), and sample drying for SEM imaging (Fig. 1). Note however that depending on conditions the distribution of morphologies may vary slightly consistent with earlier reporting of electrostatic effect for Aβ-(1-40)[15] (see Supplementary Fig. 5f). Extending beyond the diffraction limit suggests that protein spherulite growth may diverge from isotropically grown in space[7,27], and proceed in a preferential direction.

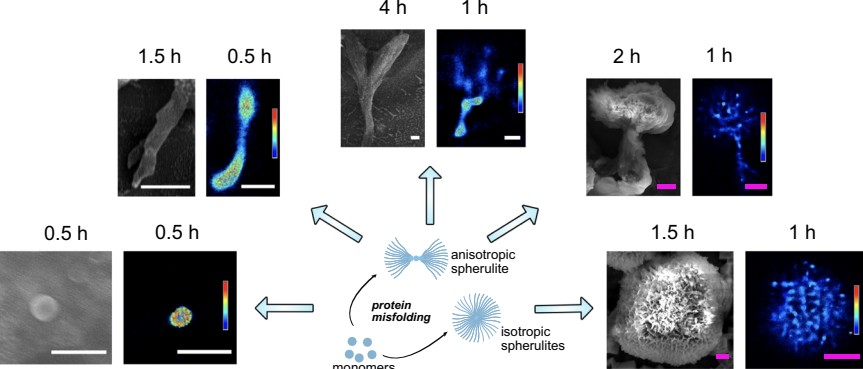

**Fig. 1 SEM and super-resolution 3D dSTORM reconstructed images of the co-existing in solution morphologies of the anisotropically/isotropically grown structures of different HI aggregates.** 3D dSTORM images: density plots, pseudocolor scale corresponds to neighbor localizations: the density of neighboring events within a 100 nm radius sphere from localization. Pseudocolor scale ranges from 0 to 1000 for the first image on left and 0 to 400 for the rest. Scale bars in white color are 1 μm and scale bars in purple color are 5 μm.

The density plots created with 3D dSTORM (Fig. 1) clearly showed that the core had a much higher density than the branching parts, consistent with previous suggestions of the existence of a low-density corona in spherulite structures[7,41]. The formation of the high-density cores appears to indicate the nucleation point, with the subsequent linear-like elongation and branching of slender threadlike fibrils resembling crystalline growth[42,43]. This is consistent with the recently proposed initial protein condensation process[44], and further growth is determined by tight fibril packing, which forces the biomolecular assembly to occur anisotropically along one specific direction. Delineating this however would require additional experiments and is beyond the scope of this study. The directly observed anisotropy challenges the isotropic spherulite growth, for which the process occurs via the formation of a radiating array of fiber crystallites, but it is observed in the case of crystalline-coil block copolymer spherulites[45]. The origin of such anisotropy might be due to the occurrence of secondary and heterogeneous nucleation at the aggregate surface[20,46], with different binding efficiencies depending on the aggregate areas. While the data in Fig. 1 would be consistent with the secondary nucleation, deciphering this with additional data falls beyond the scope of this work.

The diameter of the early linear aggregates increased as a function of time (Supplementary Table 1). This indicates that the growth was not limited to the end-to-end attachment to the linear aggregate, and lateral aggregation also took place. While this is to a certain extent expected[20], the super resolution recordings allowed its quantification. The early central linear structures, with diameters of 400 ± 100 nm (see Supplementary Table 1), successively branched to form radially oriented amyloid fiber-like structures. The further away from the core, the higher the increase in branching frequency, yielding more space-filling patterns. The dimensions of the corona-like structure were ~2 μm to >20 μm, as shown in Fig. 1.

To exclude that diverse morphologies originate from electrostatic interactions with surface immobilization[15,47], we used spinning disk microscopy and SEM to detect the morphology of spherulites at different growth stages in solution (see Supplementary Fig. 5). Consistent with the data displayed in Fig. 1, we detect both spherulites with asymmetrically grown (Fig. 2a, b) and symmetrically grown (Fig. 2c, d) (see 3D videos of Fig. 2a, c in Supplementary Movies 1 and 2). We confirmed that the asymmetric growth was not an artifact of substrate depletion, as spherulites with asymmetric lobes had already formed by 2 h of incubation (Fig. 1). This suggests that growth periods of multiple rates occurred within a single sample (Supplementary Fig. 5),

which may be masked in bulk kinetics. Moreover, our data indicated the possibility that growth did not occur entirely isotropically from the central core, but rather, there was initially a preferential direction.

**REPLOM: A super resolution method for the real time direct observation of the growth of protein aggregation.** We developed a new super-resolution experimental method based on single-molecule localization microscopy, to quantitatively measure the growth rates at the single-aggregate level while simultaneously monitoring the morphological development of the structure. We named the method REPLOM, as it allows researchers to directly image the morphological development of each individual aggregate in real-time with super-resolution and, simultaneously, access the kinetic traces for thermodynamic analysis of the process. To perform REPLOM, HI monomers were covalently labeled with Alexa Fluor 647 NHS Ester (see "Methods" 'REPLOM' section for experimental details). Figure 3a illustrates how REPLOM works: initially, only small protein condensates, i.e., cores, are formed and bind to the poly-L-lysine-covered surface. The spatial location of each of the fluorophores is accurately detected prior to their photobleaching[48,49]. Optimizing the imaging settings and the absence of imaging buffer ensures rapid chromophore bleaching after binding (see "Methods" 'REPLOM' section and Supplementary Fig. 6). As the growth progresses, additional HI monomers from the solution bind to the core, extending the dimensions of the aggregate. Each labeled insulin binding event results in a diffraction-limited spot, the precise location of which can be accurately extracted, similarly to in photoactivated localization microscopy (PALM) methodologies[50] (see "Methods" 'REPLOM' section and Supplementary Movies 3, 4). The resolution of REPLOM was determined by the FWHM of multiple spots' intensity, which was ~ 66 nm (Supplementary Fig. 7).

Parallelized recordings of the spatially distinct binding of multiple individual HI loaded with emitters allow the real-time direct observation of the temporal morphological development of each aggregate (see Fig. 3b, Supplementary Fig. 8, and Supplementary Movies 3, 4). Due to the slow kinetics of spherulite formation, the waiting time between each frame was 20–40 s, which allowed us to capture both seed formation and extract the growth rate of insulin aggregates. To investigate the effect of excitation light on protein aggregation[51], we performed a control experiment with a waiting time of 2 min between each frame at 45 °C (Supplementary Fig. 9 and Supplementary

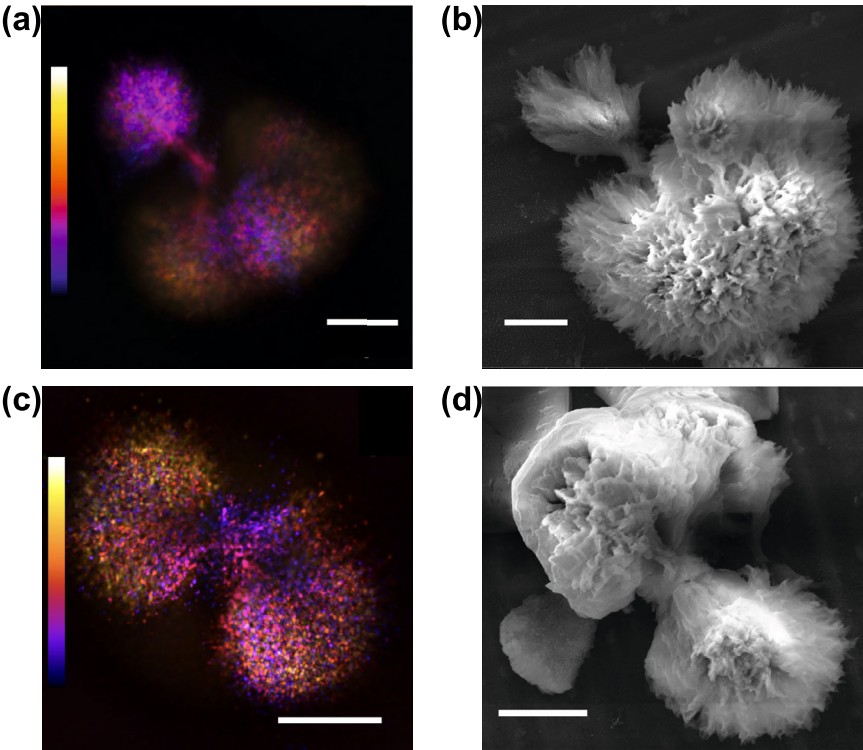

**Fig. 2 Structure of anisotropically grown human insulin spherulites of two distinct growth-morphologies. a, b** Spherulites with two asymmetric sides captured by spinning disk confocal microscopy and SEM, respectively. **c, d** Spherulites with symmetric two side structures captured by spinning disk confocal microscopy and SEM, respectively. Data in **a, c** were acquired for a sample from incubation time of 16 h at 60 °C. Data in (**b, d**) are for a sample from an incubation time of 4 h at 60 °C. Color scales are from −14.04 to 14.04 μm in **a** and −8.46 to 8.46 μm in **c**. Scale bars are 10 μm. All samples were covalently labeled with Alexa Fluor 647.

Movie 5). While the sample was 5 times less exposed to light, its growth kinetic and final structure were practically the same. This together with the results of SEM, spinning disk microscopy (Fig. 2) and TIRF images of HI spherulites with ThT as the fluorophore (Supplementary Fig. 3) has confirmed that excitation light has no effect on insulin aggregation. This could be due to the low labeled to unlabeled insulin ratio (1/10,000) and the very limited exposure to excitation light (30 ms every 20–40 s), resulting in a few hundred frames per experiment. Faster frame rates such as 20 ms are possible for the fast grown aggregates. The methodology is reliant on the intrinsic bleaching of chromophores to extract their coordinates[52–54] and is similar to binding activation localization microscopy (BALM)[55], which measures existing structures, but additionally facilitates real-time direct observation of the growth process. We also used conventional diffraction limited TIRF with 3 μM ThT as the chromophore to record the growth process of HI spherulites (Supplementary Movie 6). While the aggregation process is visible the fine structural details that define the aggregate morphology and rates are masked with diffraction-limited imaging. This further verifies the advantages of REPLOM, which extends beyond recent methods based on conventional TIRF to observe exclusively fibril growth[20] or low temporal resolution time lapses of linear or spherulite growth[15,19,21], offering in addition rate recording and morphological development of both fibrillar and spherulite structures even below the diffraction limit. Consequently, the geometry and morphological development of each aggregate can be observed directly with sub-diffraction resolution, offering the extraction of each particle's growth kinetics.

**Extraction of growth rates for diverse aggregate morphologies.** Consistent with the 3D dSTORM data, the direct observation of

HI spherulite growth by REPLOM confirmed that HI spherulites grow both anisotropically and isotropically (Fig. 3). To extract the growth rate kinetics for each individual aggregate, we identified the points belonging to the growing aggregate with an approximate Euclidean Minimum Spanning tree segmentation[56] and estimated the area using a Gaussian mixture model based on hierarchical clustering in Fig. 3c, d (see "Methods" 'Quantification of growth kinetics by Euclidean Minimum Spanning tree' section for details)[30–33]. Compared to intensity counting areas, the areas extracted by super resolution are much more precise, allowing us to resolve the multiple growth states nature within a single aggregate (Supplementary Fig. 10). For isotropic morphologies, a single linear growth rate was observed ($r_x$) followed by a plateau, while for anisotropic morphologies the growth curve consisted of two rate components ($r_1$ and $r_2$), as shown in Fig. 3c, d and Supplementary Fig. 11; $r_1$ corresponds to the initial linear core of the dendrite and $r_2$ to the branching part, and they best fitted to reaction-limited linear growth and a diffusion-limited sigmoidal growth, respectively[14,16,57–59] (see Supplementary Movies 3, 4, 7–10). Consequently, this allowed us to deconvolute the two growth rates of the anisotropically grown spherulites: the rate of the dendrite fibril like structure ($r_1$), and the rate of the branching part for each anisotropically grown spherulite ($r_2$). Similarly, the ($r_x$) for each individually isotropically grown spherulite were extracted. It worths noticing that the growth rates are the rates of growth of the radius, which reflects the 3D nature of the growth (Supplementary Fig. 12). Similarly, sudden deposition of smaller structures does not appear to bias the extracted growth rates (see Supplementary Movie 4 and Supplementary Movie 8 and the corresponding traces in Fig. 3d and Supplementary Fig. 11b respectively). The growth readouts of the individual geometrically distinct morphologies allowed us to go beyond the standard

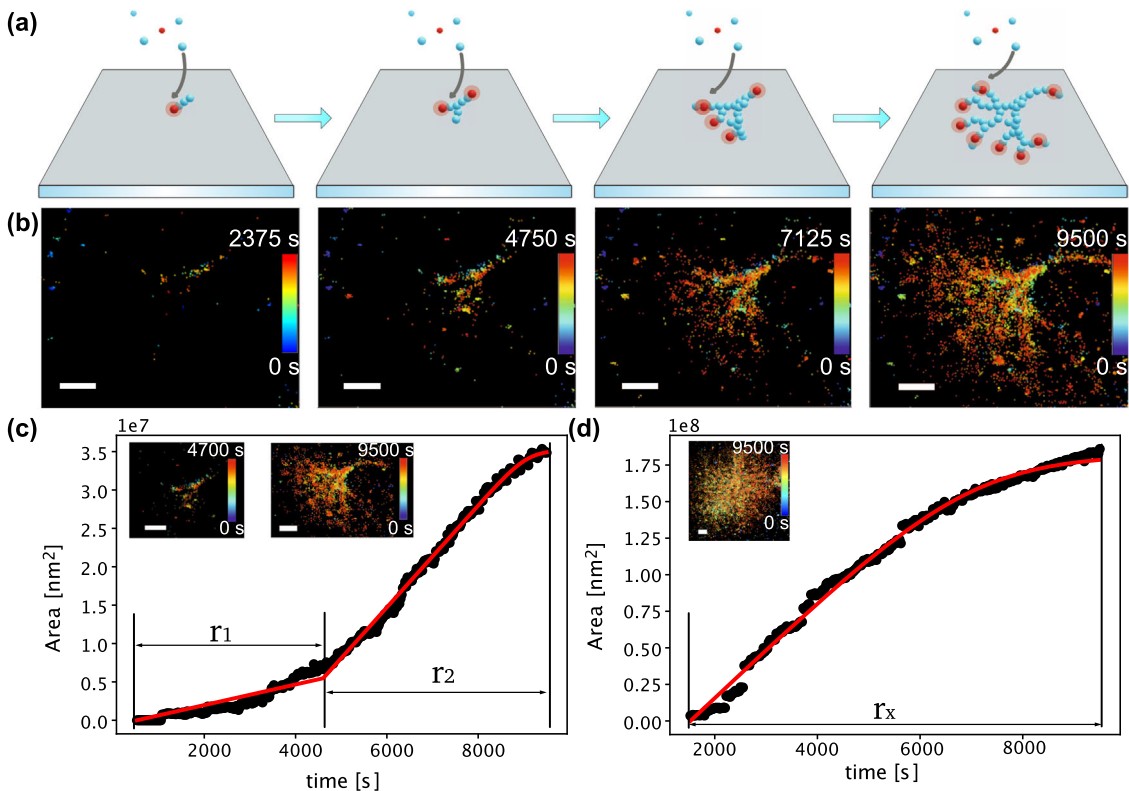

**Fig. 3 Direct real-time observation of HI aggregate growth by REPLOM (real-time kinetics via binding and photobleaching localization microscopy).**
**a** Cartoon representation of REPLOM: initially, the fluorescent signal from the small fluorescently labeled protein condensates was detected, followed by chromophore photobleaching. As the growth progressed, labeled insulins from solution bound to the aggregate, increasing the dimensions. Each binding event resulted in a diffraction-limited spot, the coordinates of which were accurately extracted, before it was photobleached by the intense laser. Parallelized recordings of the spatially distinct binding of multiple individual emitters revealed the temporal morphological development of several aggregates (Red is Alexa Fluor 647-labeled HI in fluorescent state, and blue is un-labeled insulin or Alexa Fluor 647-labeled HI in dark/photobleached state). **b** Direct real-time observation of temporal development of anisotropic growth at $t = 2375$ time intervals. Scale bars: 2 µm. Growth curves of anisotropic spherulite (**c**) and isotropic spherulite (**d**). For anisotropic spherulites, the curve contains two parts roughly correlating with the formation of the core/linear part and branching part (see "Methods" 'REPLOM' section). Isotropic spherulite growth was linear and followed by saturation. Inset: the corresponding HI spherulite obtained by REPLOM. Scale bars: 2 µm. See SI for the movies.

analysis of sigmoidal curves, which does not yield information on, or discriminate between, the temporal developments for each morphology. REPLOM revealed that the anisotropic growth operated via a two-step process imposed by the geometry of the growth—a pattern masked in current super-resolution and bulk readouts.

**Extraction of energy barriers for the growth of diverse HI spherulites morphologies**. The real-time single-particle readout from REPLOM facilitates the kinetic analysis of the temperature dependence of growth for each diffraction limited type of spherulite morphology and, consequently, the extraction of the activation energy barriers for both the spherulite morphologies and growth phase. Therefore, HI aggregate formation was induced at three different temperatures accessible without introducing optical artifacts in our microscopy setup: 45, 37, and 32 °C. The ThT fluorescence measurements at the three temperatures representing the average growth kinetics are shown in Fig. 4a. The rate distributions at the three temperatures for each type of morphological growth are shown in Fig. 4b–d ($N = ~20$, see also Supplementary Fig. 13). As expected, the linear parts $r_1$ (Fig. 4b) and branched parts $r_2$ (Fig. 4c), as well as the isotropic growth rate $r_x$ (Fig. 4d), increased at increased incubation temperature. The data do not show a pronounced curvature, and this may be due to the narrow temperature range investigated in our study and is in agreement with earlier studies[60]. This would

suggest that the differences in heat capacity between the soluble states of the proteins and the transition states for aggregation are small[60]. Using the Arrhenius equation[60,61] (Fig. 4e–h), we extracted the activation energy of each of the isotropic or anisotropic morphological growths and the respective linear or branching parts of the individual aggregates. For the linear part of the anisotropic spherulites, the activation energy was $105 \pm 10$ kJ/mol (Fig. 4f), while for the branched part it was $84 \pm 5$ kJ/mol (Fig. 4g), and for the isotropically grown spherulites it was $87 \pm 6$ kJ/mol (Fig. 4h). The lower activation energy for the highly branched part of the spherulite growth, when compared to the initial highly directional growth, may be ascribed to a change in the nature of reacting species. Indeed, the high temperature or low pH shifts the equilibrium from the ensemble of native states to an ensemble of unfolded and aggregation-prone states. This shift is likely more pronounced at later phases of the reaction (i.e., when reaching the branching phase), leading to the presence of more aggregation-prone reacting species. This will result in an overall lower activation barrier of the branched part compared to the linear part of the growth. The activation energy extracted from the bulk kinetics shown in Fig. 4e ($94 \pm 15$ kJ/mol) is consistent with data on bovine insulin fibril formation (~100 kJ/mol)[60]. The REPLOM methodology, on the other hand, allowed deconvolution of a higher barrier related to step 1 in the anisotropic growth ($r_1$) and lower barrier in the branching part of isotropic and anisotropic growth ($r_x$ and $r_2$). Together, these data

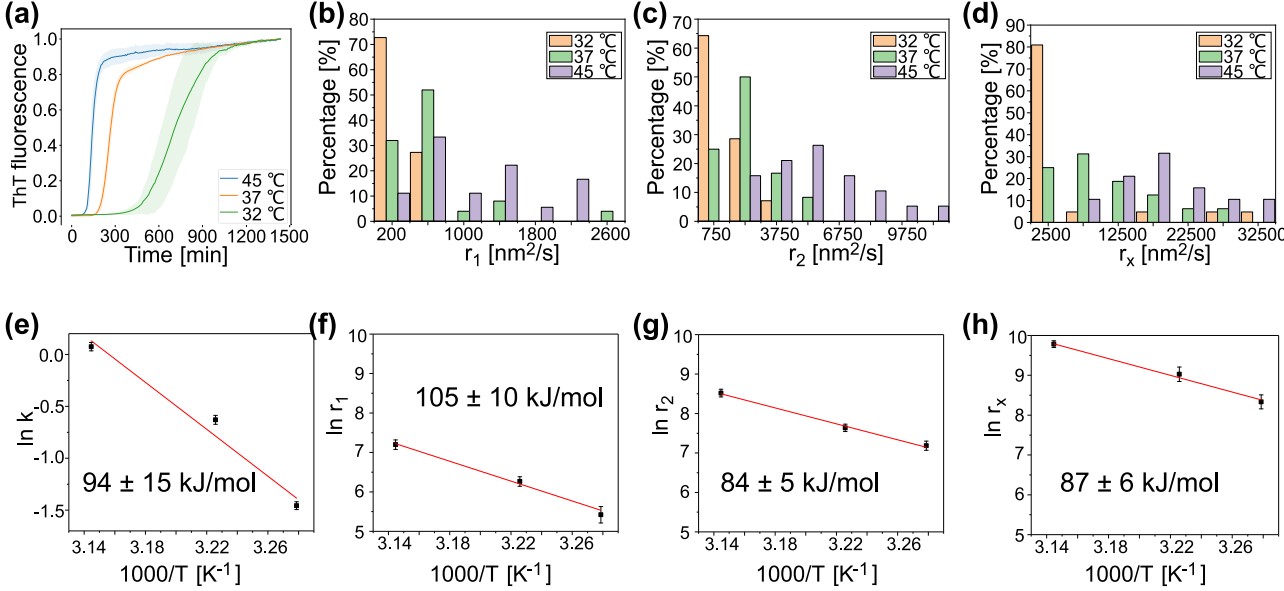

**Fig. 4 Kinetic and thermodynamic characterization of insulin aggregation. a** Normalized bulk ThT fluorescence kinetics with shaded lines of standard deviation with incubation temperatures of 45, 37, and 32 °C. $n = 4$ for each temperature. **b, c** REPLOM-extracted rate distribution of anisotropic aggregates at the three different incubation temperatures: **b** linear part ($n = 11$, 25, and 18 for 32, 37, and 45 °C, respectively) and **c** branching part ($n = 14$, 24 and 19 for 32, 37 and 45 °C, respectively). **d** Rate distribution of isotropic aggregates at the three different incubation temperatures (($n = 22$, 16, and 19 for 32, 37, and 45 °C, respectively)). Arrhenius plots for spherulites obtained from bulk experiments (**e**), and REPLOM (**f–h**). The error bars in **e–h** present standard error. The formation of linear (**f**) and branched (**g**) parts of anisotropic spherulites, and the formation of isotropic spherulites (**h**).

indicate that the pronounced heterogeneity of growth mechanisms and structures within the aggregation ensemble leads to heterogeneity of the activation barriers. Supplementary Table 2 shows the relative abundance of the types of spherulites at each incubation temperature. While one has to be careful in accessing the abundance of morphologies in such heterogeneous samples and the abundance of linear aggregates (fibrils and fibril-like linear core) are too low for quantitative assessment (4–9 structure in ~20 fields of view), we found small variations in the percentages of isotropic and anisotropic spherulites between 45 and 37 °C. Lowering the temperature to 32 °C results in an increase in the abundance of isotropic spherulites from 33 to 45%. This is consistent with isotropic spherulites with lower energy barriers being more favorable to form at low incubation temperatures. We indeed highlighted that spherulite growth may proceed both isotropically and anisotropically, with the latter presenting a two-step process imposed by the geometry of the growth and characterized by two activation energies that are markedly different to those obtained by bulk kinetics and for insulin fibrils[60].

## Conclusions

Our combined results revealed that the growth of amyloid core-shell structures for insulin, i.e., spherulites, may proceed not only via isotropic growth but also by following a multistep pathway characterized by initial pronounced anisotropic behavior (Fig. 5). The anisotropic growth may thus not be an exclusive property of metal alloys, salts, and minerals, but may extend to protein aggregates. In essence, our data are consistent with a unifying mechanism underlying chemical growth of both biological soft materials and hard-non biological composites. Such variability in growth within the same aggregation reaction results in a spectrum of aggregation kinetics traces that can be quantitatively detected by our method, allowing the operator to extract the thermodynamic parameters for each of the aggregation subsets. These findings underscore how conclusions solely based on bulk kinetics data may overlook the complexity and heterogeneity of the aggregation process.

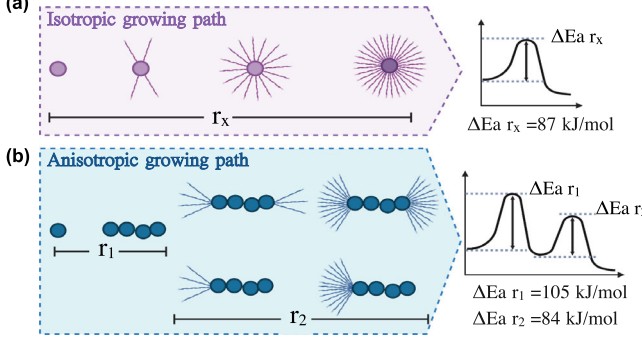

**Fig. 5 Schematic representation of the diverse pathways of insulin aggregation and their respective energy barriers. a** Isotropic spherulite growth, where fibril-like filaments isotropically and radially grow on a dense core. Process is characterized by a single activation energy of 87 kJ/mol. **b** Anisotropic growth, where the dense core is growing linearly before it successively branches to form radially oriented amyloid fiber-like structures. The further the branching from the core, the more increased the branching frequency, yielding a more space-filling pattern. The process involves two steps imposed by the geometry of the growth and characterized by two activation energies of 105 and 84 kJ/mol for the linear and branching parts, respectively.

Our experimental approach offers real-time detection of super-resolution images during protein aggregation kinetics. The REPLOM method allows the direct observation of self-assembly kinetics at the level of single aggregates and the quantification of the heterogeneity of aggregates and their growth mechanisms, which are otherwise masked by current methodologies. Our general framework can be extended to the simultaneous detection of markedly different structures within a single aggregation reaction and contribute to research into a more comprehensive representation of the generalized energy landscape of proteins. This will offer the unique possibility of disentangling different

mechanisms leading to the myriad of aggregate structures that occur. The method is implemented on the insulin model system, but can be easily translatable to more medically relevant proteins, such as α-synuclein or Aβ peptide. Deciphering whether these structures persist in the context of the cellular environment and the direct physiological implications of anisotropically grown morphologies would require a combination of our methodologies with DNA-paint and antibodies as recently developed[62]. Our approach may indeed provide information on transient intermediate species, which are nowadays recognized as the cause of progression in many diseases, in terms of both energetics and morphology. Finally, our approach is general and may be applicable to generic self-assembly reactions of systems characterized by a high degree of heterogeneity.

## Methods

**Human insulin (HI) labeling and spherulite preparation.** Alexa Fluor 647 NHS Ester (ThermoFisher Scientific) was dissolved in anhydrous-DMSO to a concentration of 2 mg/mL. 5 μL of the dye solution was added to 1 mL 5 mg/mL HI (91077C, Sigma-Aldrich, 95%) monomer solution, mixed gently and thoroughly. The mixed solution was allowed to react for ~2 h at room temperature to complete the conjugation. After that the labeled protein was purified from the excess of free dye by a PD SpinTrap G-25 column (GE Healthcare), divided into aliquots, and stored at −80 °C.

HI spherulites were formed in 0.5 M NaCl, 20% acetic acid (VWR Chemicals, 98%) solution with pH around 1.7. The ratio of labeled to unlabeled HI monomer was about 1–60,000 (dSTORM) or 1–10,000 (REPLOM), with the final concentration of HI was 5 mg/mL. The solution was filtered through 0.22 μm filters (LABSOLUTE) and then incubated in a block heater.

**Turbidity/Thioflavin T (ThT) fluorescence kinetics.** For in situ absorbance or ThT fluorescence, experiments were carried out using a plate reader system (BMG LABTECH, CLARIOstar) with 96-microwell polystyrene plates (Nalge Nunc, ThermoFisher Scientific). Each well contained 200 μL solution. The plates were covered with a self-adhesive sealing film (nerbe plus, for absorbance) or a clear polyolefin film with sealing tape (Thermo Fisher Scientific, for ThT fluorescence) to avoid evaporation of the samples and incubated at the desired temperatures without mechanical shaking. For absorbance, the excitation wavelength was 480 nm; and for ThT fluorescence, the solution contained 20 μM ThT and the emission intensity at 486 nm was recorded upon excitation at 450 nm. The signal was detected every 309 s.

**Spinning disk microscopy.** The 3D images of grown spherulites were taken by a SpinSR10-spinning disk confocal super resolution microscope (Olympus) using a silicone oil-immersion 100× objective (UPLSAPO100XS, NA = 1.35, Olympus). The Alexa Fluor 647-labeled HI spherulites were excited with a 640 nm laser (OBIS COHERENT). The exposure time was 50 ms and the z step length was 0.36 μm.

**Scanning electron microscopy (SEM).** SEM images of spherulites were taken by using a Quanta FEG 200 ESEM microscope.

**Cross polarized microscopy.** Images were collected using a 10× objective and crossed polarized which enabled spherulites to show the characteristic Maltese cross (Zeiss Axioplan Optical Microscope, Carl Zeiss).

**Total internal reflection fluorescence (TIRF) microscopy.** Diffraction limited imaging and Super resolution imaging were attained on an inverted TIRF microscope (Olympus IX-83) with a temperature adjustable 100× oil immersion objective (UAPON 100XOTIRF, NA = 1.49, Olympus). ThT and Alexa Fluor 647 were excited by a 488 nm solid state laser line and a 640 nm solid state laser line (Olympus) respectively, and reflected to a quad band filter cube (dichroic mirrors ZT640rdc, ZT488rdc and ZT532rdc for splitting and with single-band bandpass filters FF02-482/18-25, FF01-532/3-25 and FF01-640/14-25). Signal was detected by an EMCCD camera (imagEM X2, Hamamatsu).

Positively charged poly-L-lysine surfaces[63] were used to immobile insulin aggregates. A control experiment was performed to verify there was no molecular diffusion on the poly-L-lysine surface: Atto 655 labeled insulin hexamers were added on the poly-L-lysine surface, and their mobility was observed by the TIRF microscope. Thanks to the high stability of Atto 655, we can observe the insulin hexamers for a long time. It can be seen insulin hexamers were immobile on the surface without molecular diffusion (see Supplementary Movie 11).

The PSF of our TIRF microscopy was determined by the FWHM of Alexa Fluor 647 labeled insulin imaged under otherwise identical conditions. In detail, Alexa Fluor 647 labeled monomeric insulin was first filtered through a 0.22 μm filter, then added on the poly-L-lysine surface at a low density to ensure a minimal signal

overlap. The average FWHM was measured to be about 383 nm, slightly (30%) higher than the calculated under optimal conditions.

**3D direct stochastic optical reconstruction microscopy (3D dSTORM) and image analysis.** 3D dSTORM imaging was achieved by installing a cylindrical lens ($f = 500$ mm) in the emission pathway to introduce the astigmatism of point spread function (PSF)[35]. All the dSTORM imaging experiments were performed at room temperature (~21 °C). The exposure time was 30 ms, the laser power was 10–12% and 10,000 frames for each movie.

To extract $z$ information from the widths of single molecule images, we generated a calibration curve of PSF width in the lateral plane ($W_x$ and $W_y$) as a function of height by measuring Atto 655-labeled liposomes using TIRF with a step size of 10 nm and exposure time of 30 ms (see Supplementary Methods for the preparation of Atto 655-labeled liposomes). The calibration curve is shown in Supplementary Fig. 14.

The HI aggregates which were incubated in a block heater for 0.5–2 h at 60 °C. At the desired time they were added to the poly-L-Lysine treated microscope chamber[63] and incubated for 10 min at room temperature to ensure immobilization. Extra sample was washed away with Milli-Q water. Imaging buffer containing 50 mM Tris, 10 mM NaCl, 10% (w/v) glucose, 0.5 mg/mL glucose oxidase, 40 μg/mL catalase and 0.1 M MEA[64] was flushed into the chamber for dSTORM imaging. All measurements were carried out at room temperature. We tried different labeling ratios which were 1–1000, 1–10,000, and 1–60,000, and found the optimal ratio of labeled to unlabeled insulin that provided reliable signal without affecting the aggregation process or compromising resolution was 1–60,000. This is quite different from earlier dSTORM imaging of fibrils using a ratio of 1/20[24] because of the much higher 3D density of spherulites that prevented reliable super resolution imaging at high labeling ratios.

The 3D dSTORM data was analyzed by ThunderSTORM[65]. The z information of individual localizations was extracted based on the calibration curves (calculated by ThunderSTORM, shown in Supplementary Fig. 14). The detected localizations were further filtered according to their intensity and drift correction, in order to remove some possible false positive or poor quality detections. 3D super-resolution images were visualized with ViSP software[66].

**REal-time kinetic via photobleaching localization microscopy (REPLOM).** The solution containing 5 mg/mL HI monomer was first incubated in a block heater to skip the lag phase. The optimal pre-incubation time for spherulite formation on the microscope surface was found to be ~8 h for 45 °C, 20 h for 37 °C, and 75 h for 32 °C, respectively. Then they were transferred to poly-L-lysine coated glass slide chambers and covered by a lip to prevent buffer (0.5 M NaCl, 20% acetic acid, pH 1.7) evaporation during imaging (Supplementary Fig. 15).

REPLOM was performed on the same TIRF microscope setup as the 3D dSTORM without the cylindrical lens. Alexa Fluor 647 labeled HI was excited by 640 nm solid state laser lines (Olympus). We found the optimal ratio of labeled to unlabeled insulin for REPLOM to be ca. 1–10,000. A high labeling density would result in proximate fluorophores from the newly grown area emitting simultaneously and therefore cause mislocalization[24]. Too low labeling ratio may cause some details, e.g., small branching part, during spherulites growth to be undetected. Imaging was performed with ~35% laser power, an exposure time of 30 ms followed by a waiting time for each frame of 20–40 s so as to capture in real time the slow kinetics of spherulite formation. This frame rate allowed to capture both seed formation and extract the growth rate of insulin aggregates. Faster frame rates may be required for different protein aggregates[67]. All image acquisition was performed at the same incubation temperatures as in the block heater. The incubation temperatures (45, 37, and 32 °C, respectively) during the imaging processes were achieved by a heating unit 2000 (PECON).

To compare with REPLOM, we recorded spherulites growth using diffraction limited imaging (Supplementary Movie 6). The growth condition of spherulites such as HI concentration, buffer and incubation temperature were the same as REPLOM: 5 mg/mL HI monomer was mixed with 3 μM ThT in acidic buffer (0.5 M NaCl, 20% acetic acid, pH 1.7), and preincubated in a block heater at 45 °C for ~8 h to skip the lag phase. Then the solution was transferred to poly-L-lysine coated glass slide chambers and the growth processes were observed by our TIRF microscope. ThT was excited by 488 nm solid state laser line (12% laser power) and reflected to a quad band filter cube (dichroic mirrors ZT640rdc, ZT488rdc and ZT532rdc for splitting and with single-band bandpass filters FF02-482/18-25, FF01-532/3-25 and FF01-640/14-25). The waiting time between each frame was 25 s.

The data was analyzed by ThunderSTORM. Some possible false positive or poor-quality detections were removed by intensity filter. Supplementary Figure 16 shows the comparison of images prior to and after drift correction. The reconstructed images with time series were obtained by ViSP[66] software. For Quantification of growth kinetics is available in Supporting Information.

The lifetime of fluorophores in REPLOM was evaluated by checking the duration time of fluorescent state before they were photobleached. We checked 1885 individual Alexa Fluor 647 fluorophores and found they were photobleached very fast without imaging buffer (Supplementary Fig. 6). The lifetime is about $0.7845 \pm 0.0017$ frames.

The resolution of REPLOM was determined by the FWHM of single spot's intensity (Supplementary Fig. 7) using an adapted version of previously published software[49,68]. Briefly, using our subpixel resolution software, we were able to extract multiple (91) single spots (see Supplementary Fig. 7) and align all to the same center. Fitting a two-dimensional gaussian to the resulting stacked clusters allowed the reliable extraction of FWHM used to determine the obtained resolution. Using a maximum likelihood fitting scheme avoided potential bias from data binning.

**Quantification of growth kinetics by Euclidean minimum spanning tree**. The method for identification of candidates for fluorophores docking on a growing aggregate was inspired by recent published work[56] and done in the following way:

First, using all detected REPLOM spots from the movie, an approximate Euclidean Minimum Spanning tree was constructed using only the 30 nearest neighbors as candidates for edges. Regions of aggregate candidates were cut from each other by removing all edges with lengths more than the 95th percentile. This is an effective way of separating high-density regions from low-density regions. The computation was done using the function Hierarchical Clustering from the AstroML python package. Since we were interested mostly in the large insulin aggregates where the internal structure was visible, it was decided that all clusters obtained in this manner with less than 100 detected fluorophores were excluded from the subsequent analysis.

The time-dependency of the aggregate growth was found by a similar approach. At each frame, for a cluster, a refined grouping was done by cutting an approximate Euclidean Minimum Spanning tree made using 10 neighbors with a distance cutoff of 400 nm which was found to be optimal for removal of most spots outside the aggregate while still not cutting up the main group. The points from the largest subgroup resulting from this analysis were defined to be the aggregate for that frame.

The area of the aggregate was estimated using a gaussian mixture model with a component for every 5 points in the aggregate, but not less than 25 components[56]. We defined the area of the aggregate as the region lying above the average probability density in this fit. The growth profile resulting from our approach had a few artifacts like jumps and fluctuations due to mixture model fitting and aggregate segmentation, but we found that the resulting growth curve in most cases had an identifiable trend, and the results were quite consistent across parameter choices.

From the estimated area of the aggregate in each frame, a growth curve could be plotted.

The radial growth rate of such aggregates has previously been found to be either reaction-limited or diffusion-limited, leading to linear increase in time or increase as $\propto \sqrt{t}$ respectively[14,57–59]. If we assume that the estimated area of the aggregated is directly related to the radius as $A \propto R^2$ the two growth types lead to the following models

$$\frac{dA(t)}{dt} = r_1 \tag{1}$$

$$\frac{dA(t)}{dt} = \frac{1}{2}r_1 t \tag{2}$$

where the first model is diffusion limited and the second is reaction limited. We found that many of the structures where initially consistent with reaction limited diffusion and then shifted to either diffusion or reaction limited growth with a new rate. To allow for this shift, we let the growth be diffusion limited up to a switch-point $t_0$ after which the growth rate changes. We formulate one such model which ends reaction limited and one which remains diffusion limited

$$\frac{dA(t)}{dt} = \begin{cases} r_1, & t_0 > t \\ r_2, & t_0 \le t \end{cases} \tag{3}$$

$$\frac{dA(t)}{dt} = \begin{cases} r_1, & t_0 > t \\ \frac{1}{2}r_2 t, & t_0 \le t \end{cases} \tag{4}$$

Finally, without continuous flow of constituent monomer, the growth inevitably saturates at a plateau[16]. For both models, we, therefore, introduce a switch time $t_1$ after which the growth slowly saturates sigmoidally over a time interval $5\tau$

$$\frac{dA_{lin}(t)}{dt} = \begin{cases} r_1, & t_0 > t \\ r_2, & t_0 \le t < t_1 \\ r_2 \frac{1}{1+e^{\frac{5(t-\tau-t_1)}{\tau}}}, & t_1 \le t \end{cases} \tag{5}$$

$$\frac{dA_{par}(t)}{dt} = \begin{cases} r_1, & t_0 > t \\ \frac{1}{2}r_2 t, & t_0 \le t < t_1 \\ \frac{1}{2}r_2 \frac{t}{1+e^{\frac{5(t-\tau-t_1)}{\tau}}}, & t_1 \le t \end{cases} \tag{6}$$

where we introduced the names $A_{lin}$ and $A_{par}$ referring to the linear-like and parabolic-like shape of the two resulting growth curves. We found the anisotropic spherulites to fit best with $A_{lin}$ and the isotropic spherulites fit best with $A_{par}$.

When fitting an experimentally observed aggregate growth curve $\{A_i, t_i\}, i \in (0, N-1)$ the equations where numerically integrated from an initial timepoint

$(A_0, t_0)$ to the final timepoint $(A_{N-1}, t_{N-1})$. For each growth curve, the parameters $(r_1, r_2, t_0, t_1, \tau)$ were estimated with a chi2 fit. Each fit was run twice, the first fit was unweighted and were used to estimate the error bars using the standard deviation of the residuals. The second fit used the residuals in a weighted chi2 fit to obtain the final fit parameters for the growth curve.

**Statistics and reproducibility**. Bulk kinetic curves including turbidity and ThT fluorescence were extracted from at least 4 replicates for each condition. REPLOM experiments were repeated at least two additional times with a similar outcome.

**Reporting summary**. Further information on research design is available in the Nature Research Reporting Summary linked to this article.

## Data availability

All data sets used for figures are provided as source data in the manuscript. All source data are available at https://sid.erda.dk/sharelink/fje3exOlq2.

## Code availability

Source code and executable can be found at https://github.com/hatzakislab/REPLOM-analysis-tool.

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

## Acknowledgements

This work was funded by the Lundbeck foundation (grant R250-2017-1293 and R346-2020-1759) and the Carlsberg foundation (CF21-0659) for M.Z. Villum foundation young investigator fellowship (grant 10099), the Carlsberg foundation Distinguished Associate professor program (CF16-0797) and the NovoNordisk Center for Bio-pharmaceuticals and Biobarriers in Drug Delivery (NNF16OC0021948) for N.S.H. Villum foundation young investigator fellowship (grant 19175), the Novo Nordisk foundation (NNF16OC0021948) and Lundbeck foundation (R155-2013-14113) for V.F. China Scholarship Council (201709110108) for X.Z. Work at The Novo Nordisk Foundation Center for Protein Research (CPR) that NSH is associated with, is funded by a generous donation from the Novo Nordisk Foundation (Grant number NNF14CC0001). We thank Dr Y. Hu from Technical University of Denmark for the help with SEM imaging. N.S.H. and V.F. are members of the Integrative Structural Biology Cluster (ISBUC) at the University of Copenhagen.

## Author contributions

M.Z., N.S.H. and V.F. wrote the paper with feedback from all authors. M.Z. designed, carried out and analyzed all microscopy experiments, and prepared all samples. H.D.P. wrote the automated cluster finding and rates analysis algorithm. M.Z. and X.Z. did the ThT-fluorescence and turbidity measurements. S.S.-R.B. calculated the resolution of REPLOM and fluorophore's lifetime. L.B. and A.Z. helped with the mechanism explanation. N.S.H. conceived the project idea, in collaboration with V.F., and had the overall project management and strategy.

## Competing interests

The authors declare no competing interests.
