## [Peer Review File · Communications Biology]

Reviewers' comments:

Reviewer #1 (Remarks to the Author):

Proteins and peptides that form amyloid fibrils can do so in the form of spherulites, a dendrimer-like aggregate morphology that likely is the basis for florid amyloid plaque pathology in Alzheimer's and some forms of prion diseases and that is also prevalent in the huntingtin fibril formation. The dynamics of spherulite formation are yet poorly understood and the manuscript by Zhang et al. present a novel and intriguing application of super-resolution microscopy to analyze the dynamics of spherulite formation.

The authors utilize the limited photostability of dye molecules, which are covalently linked to insulin monomers, to track the growth of insulin amyloid spherulites by transiently imaging newly bound insulin molecules at the growth front of the aggregate. The authors have christened their approach REPLOM and compare their methodology to transient binding localization techniques, such as BALM, TAB or aPAINT, although on a technical level, it more resembles traditional dSTORM or PALM microscopy utilizing covalently coupled conventional dyes.

Notwithstanding the overabundance of acronyms in the field of single molecule fluorescence, this is a very interesting and potentially powerful application of localization imaging, which takes advantage of a shortcoming of fluorescent dyes, their limited photostability, to extract dynamic information about aggregate growth. However, it seems that the data presented in this manuscript does not take full advantage of the higher spatial resolution and that the kinetic data could also have been extracted from ThT TIRF microscopy experiments such as pioneered by the Goto lab, possibly even more easily, since the authors analyze the growth in total area of each spherulite rather than the growth of each filament. In doing so, it seems to me that the authors have not have utilized the full power of their super-resolution methodology as the measurement of the aggregate area could easily be achieved by conventional microscopy. This is even more surprising as the supplementary movies demonstrate that the method is capable of tracking the linear growth rate of dendrites in the spherulites, which could be used to extract dendrite fibril growth rates directly. I suggest that the authors incorporate a direct analysis of fibril growth rates into their analysis.

2) The authors show in their Figures 2 and movies S1 and S2 that spherulites are three-dimensional objects. However, the REPLOM image acquisition and analysis uses a 2D imaging mode. The authors need to discuss and possibly correct for how the 2D analysis of a 3D growth process distorts the kinetic analysis. Could this be the reason why their reported activation energies of ~ 85 kJ/mol are below the value measured by bulk fluorescence?

3) Growth of spherulites in solution and sudden deposition on surface as seen in movie S6 should also distort growth kinetics. How does the analysis deal with this?

4) Unlike the isotropically grown spherulites, the smaller anisotropic objects in S7 show largely red coded fluorescence from the end point of aggregation throughout the structure. Shouldn't there be a visible spatial distribution from blue to red in all morphologies of aggregates that mirrors the aggregate growth process?

5) How do the authors interpret the lower activation energy for the highly branched part of spherulite growth when compared to the initial highly directional growth?

Minor points:

6) Axis labels in many graphs are too small to be legible

7) Significant digits in E_a should match the errors, i.e. $E_a = 105 \pm 10$ kJ/mol, not 104.67 ± 10.23 kJ/mol

Reviewer #2 (Remarks to the Author):

The manuscript titled "Direct Observation of Heterogeneous Formation of Amyloid Spherulites in Real-time by Super-resolution Microscopy" by Hatzakis et al, the authors report a developed a super-resolution method – REPLOM (Real-time kinetics via binding and Photobleaching LOCALISATION Microscopy) to directly visualise and quantify the existence and abundance of insulin spherulites and monitor their heterogeneous growth kinetics. These studies reveal distinct pathways in the growth amyloid spherulites – isotropic and anisotropic. In combination with machine learning

approaches, the authors establish specific morphological transitions and determine energy barriers and the energy landscape at the level of single aggregate morphology. There is no doubt that the study is neat. However, the work lacks novelty in terms of the use of this method. Also, the photobleaching studies can be performed with diffraction-limited imaging to extract the same growth kinetics of fibril formation, so it is not clear what the role of super-resolution (3D STORM) is here. This is further highlighted by the fact that the authors use 60,000 fold more unlabelled monomer compared with fluorophore-labelled monomer, as it is only the latter that results in a signal in dSTORM imaging. Following corrections are necessary before the manuscript can be recommended for publication.

1. The authors establish the formulation of spherulites in conditions similar to that of fibril formation, what factors contribute to the formation of spherulites needs elaboration.
2. Figure S1 shows an average of four replicates, however the authors must also show the standard deviation of measurements at each point. There appears to be a shorter lag phase of insulin aggregation in the case of Alexa Flour 647 labelled monomer. An inset or another figure with rescaled x-axis between 0 and 300 min would be helpful to clarify this. It is interesting that the authors observe no difference in assembly kinetics, previous reports with similar studies have shown differential assembly behaviour in fluorophore-labelled amyloid proteins (Biophysical Journal 116, 2019, 227–238, and RSC Chem. Biol., 2021,2, 1491-1498). Perhaps the authors need to repeat the kinetics experiments with the ratio of labelled to unlabelled monomers they use in the imaging experiments.
3. It is unclear how the authors control the dSTORM and photobleaching experiments. The ~20 seconds required for photobleaching, molecular diffusion is a common phenomena that occurs and the effect of this on localisation acquired for the assembly kinetics cannot be ignored.
4. The supplementary information does not have details on the conditions (lasers, buffer systems etc) used for dSTORM imaging.

Reviewer #3 (Remarks to the Author):

The article by Zhang et al examine the kinetics of growth of amyloid spherulites using real time fluorescence microscopy. The authors have used human insulin (HI) as a model system for their study. Several imaging techniques such as superresolution optical microscopy and spinning disc confocal microscopy and SEM have been used to characterize the spherulites. Characterization of the growth of the spherulites is important due to its potential involvement in multiple protein aggregation diseases. Therefore, the work presented in this article is highly important. The authors report that the spherulites are broadly of two types, isotropic and anisotropic. Using a novel type of superresolution microscopy, viz, REPLOM and a machine learning based approach for analysis of the images, the authors extracted the rate of growth of individual spherulites and observed that the anisotropic spherulites exhibit two different rates of growth (viz, r_1 and r_2) and two different energy barriers in two distinct phases. Taken together it is well written paper that provides valuable information on the kinetics of growth of the spherulite type of aggregates of HI.

My critical comments are as follows.

- i) While the authors claim use of REPLOM enables them to “directly observe and quantify the existence and abundance of diverse aggregate morphologies of the model system insulin, below the diffraction limit and extract their heterogeneous growth kinetics”, it is not clear if any of the above-mentioned observations requires REPLOM. Given that the dimension of the spherulites is larger than several microns non-superresolution microscopy techniques, particularly, TIRFM or confocal microscopy could have given the same information regarding “diverse aggregate morphologies” and extraction of “their heterogeneous growth kinetics”. While superresolution microscopy is a valuable tool to observe the finer features, the images of the spherulites do not exhibit any remarkable feature. Superresolution images of the spherulites look practically featureless (for comparison see the SEM images). I would be interested to see a comparison between the images of the spherulites obtained using diffraction limited and superresolution imaging. I agree that the imaging technique used here is superior, demonstration of the superiority of the quality of the information obtained would be extremely helpful.
- ii) Once again for the measurement of the rates of growth (i.e., dA/dt), the area (A) could simply be measured by counting the number of pixels in the image above a certain threshold. It would be

helpful to compare two different approaches to find out if the rates extracted are more precise using the sophisticated ML based approach used here. I have a feeling that a simple plot of the intensity of the cluster as a function of time would demonstrate the similar plots as shown in Figure 3, panels C and D. A justification of applying a simpler approach is given below.

iii) While superresolution microscopy enables visualization of finer structural details, it has a serious drawback when applied to visualize growth of amyloids in real time. Goto and coworkers (Ozawa, JBC, 2009) have shown that exposure to excitation light can severely affect growth of the amyloid fibrils. They proposed that this is due to production of free radicals that damage the fibrils. It is possible that this is not a problem here since the authors are using a very low ratio of labeled to unlabeled peptides. However, it is necessary to verify the impact of intensity of the excitation light on the kinetics of growth of the spherulites. If indeed high intensity and longer exposure of excitation light required in superresolution microscopy affects the growth of the spherulites, then it may not be a suitable approach.

iv) For measurement of the rates of growth as a function of temperature (Fig. 4) is REPLM/dSTORM imaging performed at 32, 37 and 45 degree Celsius? In the methods section it is mentioned that all the dSTORM experiments have been performed at RT. If the experiments have been performed at RT then how are the rates extracted at higher temperatures? In my understanding the superresolution objective lenses are not recommended for use at high temperature such as at 45 degree Celsius.

v) The authors have observed considerable variability in the measured rate of growth of the individual spherulites. The origin of variability may be attributed to the heterogeneity of the aggregates. It may be noted that the rate of growth of a particular aggregate would also depend on the number of growing ends. Since we don't really know the number of growth competent ends in a particular aggregate is there a point of measuring/comparing rate of growth of single aggregates. The variability may arise purely from the differences in the number of growing ends.

vi) The authors describe at the beginning of the results section two types of spherulites, viz, isotropic and anisotropic. The isotropic one is supposed to be spherical and the anisotropic ones are non-symmetric. However, looking at the images none of these looks spherical to me. The image of the small spherulite in the left side of bottom panel in Figure 1 is the closest to being a sphere. Of course, some of the images appear asymmetric than others. Hence, it would be helpful if the authors can describe the criterion being used to define the two distinct classes.

vii) The authors have observed at least three different energy barriers corresponding to r_1 , r_2 and r . This is highly informative. Therefore, change of temperature should in principle alter the relative abundance of the types of spherulites observed. I would like to know if the authors have observed such differences.

Reviewer response

Reviewer #1 (Remarks to the Author):

Proteins and peptides that form amyloid fibrils can do so in the form of spherulites, a dendrimer-like aggregate morphology that likely is the basis for florid amyloid plaque pathology in Alzheimer's and some forms of prion diseases and that is also prevalent in the huntingtin fibril formation. The dynamics of spherulite formation are yet poorly understood and the manuscript by Zhang et al. **present a novel and intriguing application of super-resolution microscopy** to analyze the dynamics of spherulite formation. The authors utilize the limited photostability of dye molecules, which are covalently linked to insulin monomers, to track the growth of insulin amyloid spherulites by transiently imaging newly bound insulin molecules at the growth front of the aggregate. The authors have christened their approach REPLOM and compare their methodology to transient binding localization techniques, such as BALM, TAB or aPAINT, although on a technical level, it more resembles traditional dSTORM or PALM microscopy utilizing covalently coupled conventional dyes.

Notwithstanding the overabundance of acronyms in the field of single molecule fluorescence, **this is a very interesting and potentially powerful application of localization imaging, which takes advantage of a shortcoming of fluorescent dyes**, their limited photostability, to extract dynamic information about aggregate growth.

Response

We thank the reviewer for detailed reading of the manuscript, acknowledging its novelty and its intriguing applications and their favorable comments. Below we present our response and changes in the manuscript, fully and in detail addressing every one of the valuable comments.

1) However, it seems that the data presented in this manuscript **does not take full advantage of the higher spatial resolution** and that **the kinetic data could also have been extracted from ThT TIRF microscopy experiments** such as pioneered by the Goto lab, possibly even more easily, since the authors analyze the growth in total area of each spherulite rather than the growth of each filament. In doing so, it seems to me that the authors have not have utilized the full power of their super-resolution methodology as the measurement of the aggregate area could easily be achieved by conventional microscopy. This is even more surprising as the supplementary movies demonstrate that the method is capable of tracking the linear growth rate of dendrites in the spherulites, which could be used to extract dendrite fibril growth rates directly. I suggest that the authors incorporate a direct analysis of fibril growth rates into their analysis.

We thank the reviewer for valuable comments highlighting the required but missing elements. We are fully aware of the work of Goto lab and had cited it of course in the original manuscript for fibril growth recordings (ref 19). The referee discusses the comparison of super resolution readout of REPLOM to diffraction limited microscopy such as ThT based TIRF and the extraction of kinetic rates.

To fully address the comment on the imaging part, we provided two new experiments: We firstly compared directly the images of HI spherulites labeled either with Alexa Fluor 647 (the same chromophore as used in dSTORM) or ThT by conventional diffraction limited microscopy to 3D dSTORM (see Figure S5).

Secondly, we recorded the spherulite growth using diffraction limited ThT fluorescence as the reviewer suggested. As expected, the structural features are masked in the diffraction limited images. The ThT concentration we used is much lower than the literature ^{1,2}. Despite this, the much higher density of spherulites as compared to fibrils results in ThT fluorescence signal that is too bright for large spherulites to extract their correct structure and area information (see movie S10). Decreasing further ThT concentration would alleviate this challenge at the expense of making it practically impossible to distinguish the small protein condensates. As a result, it's hard to use ThT TIRFM to extract the kinetic data of spherulites.

The reviewer also commented on the extraction of rates for the linear fibril-like growth part. We wish to highlight here that we indeed have provided the quantification of the linear fibril-like growth rate in the manuscript. This corresponds to the rate (r_1) provided in Figure 3 and Figure 4. Acknowledging this may have not been very clear, we clarified this further in the text (see "changes in the manuscript" section below).

Changes in the manuscript

To fully address the comment, we have:

- a) Added in line 107 that "Direct comparison of 3D dSTORM recording with conventional readouts using the same chromophore or ThT fluorescence reveals that the distinct features for the aggregates are masked by diffraction limited imaging (Figure S5)".
- b) Added Figure S5 to compare the image quality of 3D dSTORM and diffraction-limited images.
- c) Added the following paragraph in line 207: "We also used conventional diffraction limited TIRF with 3 μ M ThT as the chromophore to record the growth process of HI spherulites (Supplementary Movie S10). While the aggregation process is visible the fine structural details that define the aggregate morphology and rates are masked with diffraction limited imaging. This further verifies the advantages of REPLOM".
- d) Added Supplementary Movie 10 showing the real-time growth of spherulites observed by ThT TIRFM.
- e) Added a paragraph in line 420 to describe the ThT TIRF measurement: "To compare with REPLOM, we recorded spherulites growth using diffraction limited imaging (Supplementary Movie 9). The growth condition of spherulites such as HI concentration, buffer and incubation temperature were the same as REPLOM: 5 mg/mL HI monomer was mixed with 3 μ M ThT in acidic buffer (0.5 M NaCl and 20% acetic acid, pH 1.7), and preincubated in a block heater at 45 °C for ~8h to skip the lag phase. Then the solution was transferred to poly-L-lysine coated glass slide chambers and the growth processes were observed by our TIRF microscope. ThT was excited by 488 nm solid state laser line (12% laser power) and reflected to a quad band filter cube (dichroic mirrors ZT640rdc, ZT488rdc and ZT532rdc for splitting and with single-band bandpass filters FF02-482/18-25, FF01-532/3-25 and FF01-640/14-25). The waiting time between each frame was 25 seconds."
- f) Acknowledging the growth rate fibril in the dendrite is not clear we added in line 246: "Consequently, this allowed us to deconvolute the two growth rates of the anisotropically grown spherulites: the rate of the dendrite fibril like structure (r_1), and the rate of the branching part for each anisotropically grown spherulite (r_2). Similarly, the (r_x) for each individually isotropically grown spherulite were extracted".

2) The authors show in their Figures 2 and movies S1 and S2 that spherulites are three-dimensional objects. However, the REPLOM image acquisition and analysis uses a 2D imaging mode. The authors need to discuss and possibly correct for how the 2D analysis of a 3D growth process distorts the kinetic analysis. Could this be the reason why their reported activation energies of ~ 85 kJ/mol are below the value measured by bulk fluorescence?

This is a great comment and thanks for allowing us to comment on this, here and in the manuscript. We agree that the consideration and correction for biases from 2D analysis of 3D objects in microscopy are central to studies like this one. This is why we decided to fit the growth of the measured area projection by a model which takes this into account (see Methods, "Quantification of growth kinetics by Euclidean Minimum Spanning tree" in line 495). Maybe it is not clear enough in the original manuscript and we have therefore added a supplementary figure (Figure S15) in the revised manuscript, to visualize the model along with a paragraph in the main text highlighting how the fitting function takes the 3D nature of the growth into account when fitting to obtain growth rates. We therefore don't think that it is the analysis methodology which explains the difference between bulk and single aggregate activation energies. This most likely arises from currently the ensemble average of single aggregate growth modalities with weighting factors unknown to us.

Changes in the manuscript

- a) Added Figure S15 to visualize how the fitting function reflects the 3D nature of the growth.
- b) Added a paragraph in line 249: "It worth noticing that the growth rates are the rates of growth of the radius, which reflects the 3D nature of the growth (see Methods and Figure S15).".

3) Growth of spherulites in solution and sudden deposition on surface as seen in movie S6 should also distort growth kinetics. How does the analysis deal with this?

We thank the reviewer for this very interesting comment. We are very careful about the sudden deposition of large spherulites, and we excluded them from the analysis of the kinetic rates as it may distort kinetics. The reviewer correctly noticed the sudden deposition in movie S6 which also is present in movie S4. The resulting small jump doesn't affect the growth rates (see Figure 3d and Figure S13b).

Changes in the manuscript

To fully address the comment, we have added in Line 250: "Similarly, sudden deposition of smaller structures does not appear to significantly bias the extracted growth rates (see Supplementary Movie 4 and Supplementary Movie 6, and the corresponding traces in Figure 3d and Figure S13b respectively)."

4) Unlike the isotropically grown spherulites, the smaller anisotropic objects in S7 show largely red coded fluorescence from the end point of aggregation throughout the structure. Shouldn't there be a visible spatial distribution from blue to red in all morphologies of aggregates that mirrors the aggregate growth process?

This is a good point indeed. The pseudocolor corresponds to the time of the imaging. Spherulite growth is a heterogeneous process. Many spherulites initiate their growing not exactly from the beginning of the recording. If the spherulite starts growing late, e.g. during the second half or even at the end of the recording, it may be largely red coded, like some small anisotropic aggregates in Figure S8 (as we added new supplementary figures, the Figure S7 in the original version became Figure S8 now).

Changes in the manuscript

We have added a sentence in the legend of Figure S8 to describe this: “It worth noticing that if spherulites initiate their growing during the late of the recording, there will be no blue/green pseudocolor and most of the structure will be in red pseudocolor”.

5) How do the authors interpret the lower activation energy for the highly branched part of spherulite growth when compared to the initial highly directional growth?

We thank the reviewer for the comment and for giving us the opportunity of further reflecting on this. The activation energy (i.e., activation enthalpy) for a chemical reaction depends upon the nature of reacting species and it is independent of temperature, concentration and collision frequency. In our specific system and at the level of single protein molecule, the high temperature shifts the equilibrium from the native state to an unfolded and aggregation prone state. This shift is more pronounced as the reaction proceeds (i.e. when reaching the branching), leading to more aggregation-prone reacting species and a lower activation energy compared to the one determined for the early phases.

Changes in the manuscript

In the revised manuscript, we elaborated on this aspect and added the following paragraph in line 286: “The lower activation energy for the highly branched part of the spherulite growth when compared to the initial highly directional growth may be ascribed to a change of the nature of reacting species. Indeed, the high temperature or low pH shifts the equilibrium from the ensemble of native states to an ensemble of unfolded and aggregation-prone states. This shift is likely more pronounced at later phases of the reaction (i.e., when reaching the branching phase), leading to the presence of more aggregation-prone reacting species. This will result into an overall lower activation barrier of the branched part compared to the linear part of the growth.”

Minor points:

6) Axis labels in many graphs are too small to be legible

We thank the reviewer for this comment. We have re-plotted the graphs with much larger axis labels.

Changes in the manuscript

Figure 3, Figure 4, Figure S13 and Figure S16 were replaced with larger axis labels.

7) Significant digits in E_a should match the errors, i.e. $E_a = 105 \pm 10$ kJ/mol, not 104.67 ± 10.23 kJ/mol

Thanks for noticing this. We have corrected the significant digits in all E_a (line 285, 286, 294, and Fig.5) in the revised manuscript.

Reviewer #2 (Remarks to the Author):

The manuscript titled “Direct Observation of Heterogeneous Formation of Amyloid Spherulites in Real-time by Super-resolution Microscopy” by Hatzakis et al, the authors report a developed a super-resolution method – REPLOM (Real-time kinetics via binding and Photobleaching LOcalisation Microscopy) to directly visualise and quantify the existence and abundance of insulin spherulites and monitor their heterogeneous growth kinetics. These studies reveal distinct pathways in the growth amyloid spherulites – isotropic and anisotropic. In combination with machine learning approaches, the authors establish specific

morphological transitions and determine energy barriers and the energy landscape at the level of single aggregate morphology. There is no doubt that the study is neat. However, the work lacks novelty in terms of the use of this method. Also, the photobleaching studies can be performed with diffraction-limited imaging to extract the same growth kinetics of fibril formation, so it is not clear what the role of super-resolution (3D STORM) is here. This is further highlighted by the fact that the authors use 60,000 fold more unlabelled monomer compared with fluorophore-labelled monomer, as it is only the latter that results in a signal in dSTORM imaging. Following corrections are necessary before the manuscript can be recommended for publication.

We thank the reviewer for critically reading the manuscript and recognizing its importance in deciphering the growth pathways and kinetics of spherulites. We are grateful for the critical points outlined above and the specific comments below as fully and in detail addressing them allowed us to clear some elements that needed further clarification in the manuscript.

We fully agree that diffraction-limited imaging can be used to extract the growth kinetics of fibrils and as a matter of fact, we had cited the respective literature. We respectfully disagree, however, that diffraction limited microscopy can extract growth rates of spherulites. As we have detailed below and in the revised Figure S5, S14 and supplementary movie 10 (see also comments 1 and 2 of reviewer 3) diffraction-limited imaging could provide some info on the types of spherulites (i.e. isotropic or anisotropic spherulites). However, due to the high density and complex structure of spherulites their structural details, as well as the growth kinetics are completely masked in the diffraction limited images. We verified this by recording diffraction limited imaging with Alexa Fluor 647 and ThT fluorescence and comparing to REPLOM readouts (see also the response to comment 1 of reviewer 1).

The 3D-dSTORM was used to identify the existence of diverse growth pathways and final structures of spherulites. While acquisition at different incubation times revealed the diverse structures, however provided very limited information of kinetics of growth of each structure, something that REPLOM conveniently provides. The 3D-dSTORM also confirms that the heterogeneous morphologies recorded here are not artefactual or arising from the continuous sample illumination. We wish to highlight that the high density of spherulites prevents accurate imaging with a higher labeling ratio. We have now updated this information in Methods (line 389).

Changes in the manuscript

a) Discussed in the main text (line 105) the necessity of using super-resolution imaging: "Using 3D dSTORM allowed us to extend beyond diffraction-limited imaging by TIRF microscopy, which may mask the structural details of the final spherulite structure and growth directionality³ (Figure 1 and Figure S5). Direct comparison of 3D dSTORM recording with conventional readouts using the same chromophore or ThT fluorescence, reveals that the distinct features of the aggregates are masked by diffraction limited imaging (Figure S5)."

b) Added Supplementary Movie 10 showing how the real-time growth of spherulites looks like by ThT TIRFM.

c) Added Figure S5 comparing the image quality of 3D dSTORM and diffraction-limited images, to further confirm the necessity of super-resolution imaging.

d) Added in methods in line 389 that the optimal labeling ratio of 1/60000 was utilized after testing 1/1000, 1/10000 and 1/60000: "We tried different labeling ratios which were 1 to 1000, 1 to 10000 and 1 to 60000, and found the optimal ratio of labeled to unlabeled insulin that provided reliable signal without affecting the aggregation process or compromising resolution was 1 to 60,000".

1. The authors establish the formulation of spherulites in conditions similar to that of fibril formation, what factors contribute to the formation of spherulites needs elaboration.

We thank the reviewer for the comment. The conditions for the formation of spherulites are reported in the literature ^{4,5} with a number of papers from the group of one of the senior authors Vito Foderà ^{6,7}. The conditions are identical to the ones used for the formation of fibrils ^{8,9}. Indeed the two species co-exist ⁸ and this is a further fact that justifies the need of having single-aggregate approaches to evaluate the heterogeneity of the system in a reliable way. We have demonstrated that a number of factors determine the formation of spherulites, from salt, pH and protein concentration ¹⁰, to the addition of alcohols ¹¹ and surfactants ¹². In the present work, we used well-established conditions for the formation of the aggregates. We used a fixed amount of salt and low pH and guaranteed a time scale of the process, which allows the real-time microscopy analysis. Moreover, thanks to the specific signature of the spherulites under cross-polarized microscopy, we could identify their presence by a direct observation, using this as a criterion to evaluate their presence (Figure S2).

Changes in the manuscript

In the revised version, we elaborate on this aspect and we added the following paragraph (line 65-71):
“The in vitro conditions leading to amyloid spherulites are identical to the ones used for the formation of fibrils ^{8,9}. Indeed the two species co-exist ⁸ and this is a further fact that justifies the need of having single-aggregate approaches to evaluate the heterogeneity of the system. Salt, pH and protein concentration ¹⁰, addition of alcohols ¹¹ and surfactants ¹² have been shown to contribute to the variability of the structure and morphology of spherulites. We chose the experimental conditions which guaranteed a time scale of the process allowing our real-time microscopy analysis.”

2. Figure S1 shows an average of four replicates, however the authors must also show the standard deviation of measurements at each point. There appears to be a shorter lag phase of insulin aggregation in the case of Alexa Fluor 647 labelled monomer. An inset or another figure with rescaled x-axis between 0 and 300 min would be helpful to clarify this. It is interesting that the authors observe no difference in assembly kinetics, previous reports with similar studies have shown differential assembly behaviour in fluorophore-labelled amyloid proteins (Biophysical Journal 116, 2019, 227–238, and RSC Chem. Biol., 2021,2, 1491-1498). Perhaps the authors need to repeat the kinetics experiments with the ratio of labelled to unlabelled monomers they use in the imaging experiments.

We thank the reviewer for this great comment. We have replotted all the ThT and turbidity kinetic curves with shaded lines of standard deviation at each point.

We have redone the turbidity kinetic experiments at 45 degrees with the ratio of labeled to unlabeled insulin monomers being 1/10000, the same as we used in REPLOM. Due to the low labeling ratio, Alexa Fluor 647 labeled insulin has no influence on the aggregation kinetics.

Changes in the manuscript

- a) Added standard deviation in Figure 4a and Figure S1.
- b) Replotted Figure S1b by our new obtained data and added the ratio of Alexa Fluor 647 labeled insulin.
- c) Added an inset in Figure S1b with the rescaled x axis.

d) Added a sentence in the revised manuscript (line 98): “Unlike previous studies ^{13,14}, we found fluorophore labeling had no influence on the aggregation kinetics (Figure S1b). This may be because we used quite a low ratio of labeled insulin to unlabeled insulin monomer.”

3. It is unclear how the authors control the dSTORM and photobleaching experiments. The ~20 seconds required for photobleaching, molecular diffusion is a common phenomena that occurs and the effect of this on localisation acquired for the assembly kinetics cannot be ignored.

We thank the reviewer for this comment that allowed us to clarify further both how experiments are done as well as the control experiments.

In this work, insulin aggregates were immobile on the positively charged poly-L-lysine surface. This way ensures their complete absence of diffusion. To further justify we added an extra control experiment, adding Atto 655 labeled insulin hexamers on poly-L-lysine surface in otherwise identical experimental conditions. Imaging with TIRF microscope (movie S11) reveals that they are completely immobile throughout the experimental timeframe.

The referee also comments on the control of the dSTORM and photobleaching experiments. We wish to clarify here that we used two methods. The existing 3D dSTORM and the developed REPLOM. We used imaging buffer in 3D dSTORM to prevent fluorophores from photobleaching. REPLOM on the other hand is reliant on the photobleaching of the molecules so as to prevent signal saturation and to allow diffraction limited localization of each chromophore. We therefore used a high laser power and no imaging buffer to induce fluorophores to photobleach fast (see Methods-Microscopy).

Changes in the manuscript

a) Added Supplementary Movie 11 to verify there’s no molecular diffusion in our system.

b) Added a paragraph in line 369 to describe the experiment of Supplementary Movie 11: “Positively charged poly-L-lysine surfaces¹⁵ were used to immobile insulin aggregates. A control experiment was performed to verify there was no molecular diffusion on the poly-L-lysine surface: Atto 655 labeled insulin hexamers were added on the poly-L-lysine surface, and their mobility was observed by the TIRF microscope. Thanks to the high stability of Atto 655, we can observe the insulin hexamers for a long time. It can be seen that insulin hexamers were immobile on the surface without molecular diffusion (see Supplementary Movie 11)”.

4. The supplementary information does not have details on the conditions (lasers, buffer systems etc) used for dSTORM imaging.

We thank the reviewer for noticing it. This has been rectified in the revised version.

Changes in the manuscript

Added details about experimental conditions in Methods-Microscopy section (line 364, 380, 387, 408 and 413).

Reviewer #3 (Remarks to the Author):

The article by Zhang et al examine the kinetics of growth of amyloid spherulites using real time fluorescence microscopy. The authors have used human insulin (HI) as a model system for their study. Several imaging techniques such as superresolution optical microscopy and spinning disc confocal microscopy and SEM have been used to characterize the spherulites. Characterization of the growth of the spherulites is important due to its potential involvement in multiple protein aggregation diseases. Therefore, **the work presented in this article is highly important**. The authors report that the spherulites are broadly of two types, isotropic and anisotropic. Using a novel type of superresolution microscopy, viz, REPLOM and a machine learning based approach for analysis of the images, the authors extracted the rate of growth of individual spherulites and observed that the anisotropic spherulites exhibit two different rates of growth (viz, r_1 and r_2) and two different energy barriers in two distinct phases. Taken together **it is well written paper that provides valuable information on the kinetics of growth of the spherulite type of aggregates of HI**.

We thank the reviewer for acknowledging the article is highly important and that it provides valuable information, and the valuable comments. Fully and in detail addressing all of them helped us further improve the quality of the manuscript.

My critical comments are as follows.

i) While the authors claim use of REPLOM enables them to “directly observe and quantify the existence and abundance of diverse aggregate morphologies of the model system insulin, below the diffraction limit and extract their heterogeneous growth kinetics”, it is not clear if any of the above-mentioned observations requires REPLOM.

Given that the dimension of the spherulites is larger than several microns non-superresolution microscopy techniques, particularly, TIRFM or confocal microscopy could have given the same information regarding “diverse aggregate morphologies” and extraction of “their heterogeneous growth kinetics”. While superresolution microscopy is a valuable tool to observe the finer features, the images of the spherulites do not exhibit any remarkable feature. Superresolution images of the spherulites look practically featureless (for comparison see the SEM images). I would be interested to see a comparison between the images of the spherulites obtained using diffraction limited and superresolution imaging. I agree that the imaging technique used here is superior, demonstration of the superiority of the quality of the information obtained would be extremely helpful.

Thank you for the great comment and for allowing us to more directly highlight the superiority of REPLOM as compared to diffraction limited imaging.

We imaged spherulites using diffraction limited microscopy and compared the images with 3D dSTORM, as the reviewer suggested, see Figure S5. It can be seen that the diffraction-limited imaging could provide some basic info on the types of spherulites (i.e. isotropic or anisotropic spherulites). However, their structural details are completely masked in the diffraction limited images.

To further verify this, we used ThT, the most abundantly used chromophore for protein aggregation, and recorded the growth of spherulites by conventional diffraction limited TIRF microscope (Supplementary movie 10). Due to the high density of spherulites compared to fibrils, the ThT fluorescence became too bright for large spherulites to extract their correct structure and area information. As a result, it's challenging to extract the correct underlying structure and with confidence decipher isotropic from anisotropic growth (see data in Figure S5 a and b) and practically impossible to precisely extract area information, especially for the initial stages of growth where particles are close to the diffraction limit.

Decreasing ThT concentration or lowering the laser power would minimize this challenge at the expense of making it practically impossible to distinguish the small protein condensates. As a result, it's hard to use ThT TIRFM to extract the kinetic data of spherulites.

Changes in the manuscript

To fully address the comment, we have:

a) Added a sentence in the main text (line 105) to discuss the necessity of using super-resolution imaging: "Using 3D dSTORM allowed us to extend beyond diffraction-limited imaging by TIRF microscopy, which may mask spherulite structure details and growth directionality 35 (Figure 1 and Figure S5). Direct comparison of 3D dSTORM recording with conventional readouts using the same chromophore or ThT fluorescence, reveals that the distinct features of the aggregates are masked by diffraction limited imaging (Figure S5).".

b) Added Figure S5 to compare the image quality of 3D dSTORM and diffraction-limited images.

c) Added Supplementary Movie 10 showing the real-time growth of spherulites observed by ThT TIRFM couldn't extract precise kinetics.

d) Added the following paragraph in line 207: "We also used conventional diffraction limited TIRF with 3 μ M ThT as the chromophore to record the growth process of HI spherulites (Supplementary Movie S10). While the aggregation process is visible the fine structural details that define the aggregate morphology and rates are masked with diffraction limited imaging. This further verifies the advantages of REPLOM".

e) Added a paragraph in line 420 to describe the ThT TIRF measurement: "To compare with REPLOM, we recorded spherulites growth using diffraction limited imaging (Supplementary Movie 10). The growth condition of spherulites such as HI concentration, buffer and incubation temperature were the same as REPLOM: 5 mg/mL HI monomer was mixed with 3 μ M ThT in acidic buffer (0.5 M NaCl and 20% acetic acid, pH 1.7), and preincubated in a block heater at 45 °C for ~8h to skip the lag phase. Then the solution was transferred to poly-L-lysine coated glass slide chambers and the growth processes were observed by our TIRF microscope. ThT was excited by 488 nm solid state laser line (12% laser power) and reflected to a quad band filter cube (dichroic mirrors ZT640rdc, ZT488rdc and ZT532rdc for splitting and with single-band bandpass filters FF02-482/18-25, FF01-532/3-25 and FF01-640/14-25). The waiting time between each frame was 25 seconds."

ii) Once again for the measurement of the rates of growth (i.e., dA/dt), the area (A) could simply be measured by counting the number of pixels in the image above a certain threshold. It would be helpful to compare two different approaches to find out if the rates extracted are more precise using the sophisticated ML based approach used here. I have a feeling that a simple plot of the intensity of the cluster as a function of time would demonstrate the similar plots as shown in Figure 3, panels C and D. A justification of applying a simpler approach is given below.

This is an excellent point. There are certainly regimes of the growth where we expect such an approach to work just as well as super resolution imaging. To test this, we compared our area calculation to an estimate of the resulting image if all the labeled monomers were not photobleached, both with and without the effect of a PSF (all calculations were done under the assumption of zero pixel size to investigate the optimal performance of intensity-based analysis) (Figure S14). Analysis of the intensity vs. time can be done as well with super-resolution (Figure S14, green curve) but certainly not with diffraction limited resolution. While both the super resolution fitted PSF intensities (Figure S14 green curve) and area estimation methods (Figure S14 blue and red curve) captured the two-step nature of a growing aggregate, the normal

resolution “pixel counting” reproduced only the asymptotic area growth behavior (Figure S14 grey curve). We therefore find a need for super resolution with REPLOM to measure the early nucleating morphology in an unbiased manner. We found this to be due to a merging of clusters in early growth despite the distinct branching structure present in the initial growth mode (Figure S14A). Such problems are reduced as the aggregates grew larger and the dimensions of the structure became significantly larger than the resolution of any of the methods (Figure S14 C,D), but the original morphologies would be averaged out. **Therefore, we find that REPLOM, when used to measure protein aggregation area, exceeds the resolution of currently available methods and is necessary to quantify structural transition in the early growth dynamics of the nucleating aggregate.**

Changes in the manuscript

a) Added Figure S14 to compare the differences between areas extracted by super resolution and intensity.
b) Added a small section in line 238: “Compared to intensity counting areas, the areas extracted by super resolution are much more precise, allowing us to resolve the multiple growth states nature within a single aggregate (Figure S14).”

iii) While super resolution microscopy enables visualization of finer structural details, it has a serious drawback when applied to visualize growth of amyloids in real time. Goto and coworkers (Ozawa, JBC, 2009) have shown that exposure to excitation light can severely affect growth of the amyloid fibrils. They proposed that this is due to production of free radicals that damage the fibrils. It is possible that this is not a problem here since the authors are using a very low ratio of labeled to unlabeled peptides. However, it is necessary to verify the impact of intensity of the excitation light of the kinetics of growth of the spherulites. If indeed high of intensity and longer exposure of excitation light required in super resolution microscopy affects the growth of the spherulites, then it may not be a suitable approach.

We thank the reviewer for this comment. We actually had compared the growing structures by REPLOM to the final structures recorded by 3D dSTORM, SEM, spinning disk microscopy and ThT TIRF (Figure 1, Figure 2, Figure 3, Figure S5 and Figure S6) and showed them to be practically identical. Acknowledging this may not be enough to justify the minimal effect of photodamage on our setup we used another chromophore (ThT) and a 488nm laser to observe the growth of insulin spherulites in real-time, and we can see spherulites were formed with similar incubation time (Supplementary Movie 10). We also performed REPLOM with 5 times longer waiting time of 2 minutes between each frame, instead of the original a waiting time of 25s which presented in the manuscript (Figure S12 and Figure S16). While the sample is recorded with 5x less exposure to light the final structure is practically the same supporting the minimal photodamage effect on our conclusions. All above verify the excitation light has no influence on the growth kinetics of spherulites in our conditions.

Changes in the manuscript

a) Added a sentence in line 197 of the manuscript: “To investigate the effect of excitation light on protein aggregation ¹⁶, we performed a control experiment with a waiting time of 2 minutes between each frame at 45 °C (Figure S12 and Supplementary Movie 9). While the sample was 5 times less exposure to light, its growth kinetic and final structure were practically the same. This together with the results of SEM, spinning disk microscopy (Figure 2) and TIRF images of HI spherulites with ThT as the fluorophore (Figure S5) have confirmed that excitation light has no effect on insulin aggregation.”
b) Added Figure S12 showing the kinetic of spherulites obtained by REPLOM with ~5 times longer waiting time of 2 minutes.

c) Added Supplementary Movie 9 to show the growth process of the spherulite in Figure S12.
d) Added Figure S5 to show the morphologies of spherulites incubated in a block heater and with ThT as the fluorophore have no differences from the ones obtained by REPLOM and SEM.

iv) For measurement of the rates of growth as a function of temperature (Fig. 4) is REPLOM/dSTORM imaging performed at 32, 37 and 45 degree Celsius? In the methods section it is mentioned that all the dSTORM experiments have been performed at RT. If the experiments have been performed at RT then how are the rates extracted at higher temperatures? In my understanding the super resolution objective lenses are not recommended for use at high temperature such as at 45 degree Celsius.

We thank the reviewer for the comment that allowed us to further clarify how the experiments were done. We used two super-resolution imaging methods in this work: 3D dSTORM to measure the morphologies of spherulites and REPLOM to record directly and in real time the growth of spherulites. The 3D dSTORM recordings were performed at room temperature for structures that were pre-formed at 60 degree with different incubation times. While REPLOM experiments were performed at 32, 37 and 45 degree Celsius respectively (See line 423 and 467 in the manuscript). It's true that 45 degree is not the optimal working temperature of most of the objective lenses. Temperature variations might induce chromatic aberration in objectives and improper calibration of number of photons to chromophores. The objectives used in this work (UAPON 100XOTIRF, Olympus) are apochromatically corrected and adjustable to work at diverse temperatures (from 23 to 37). At 45 degree, we adjusted the collar of the objective to find the best place, and the signal was as good as the ones at 32 and 37 degrees. Temperature Variations are not expected to bias our readouts as the final super resolved structures are fully consistent with 3D dSTORM.

Changes in the manuscript

Described our temperature adjustable objective and the temperatures used in dSTORM and REPLOM in Methods-Microscopy section (line 363, 379 and line 417 to 419).

v) The authors have observed considerable variability in the measured rate of growth of the individual spherulites. The origin of variability may be attributed to the heterogeneity of the aggregates. It may be noted that the rate of growth of a particular aggregate would also depend on the number of growing ends. Since we don't really know the number of growth competent ends in a particular aggregate is there a point of measuring/comparing rate of growth of single aggregates. The variability may arise purely from the differences in the number of growing ends.

We agree that the reasons for the distribution of the growth rates may have diverse origin, and not only related to the availability of the growing ends. The aggregation reaction is a stochastic process¹⁷ and it is intrinsically heterogeneous in space and in time¹⁸. This means that we can have an intrinsic variability in the activation time of the process in different areas of the sample. Moreover, we also expect that the rate constants will be dependent on the presence of available unreacted monomers near the nucleation sites. This amount can be highly variable and, importantly, can change over time, especially if one considers the recent development in the field indicating that liquid-liquid phase separation can occur¹⁹. The latter phenomenon can indeed create areas of low protein concentration, which will definitely affect the growth mechanism at the level of single aggregates, potentially causing what we experimentally observe. Moreover, we also expect this phenomenon to be dependent on the protein sequence. From the above facts, it is clear that unambiguously explaining the origin of this variability would require a dedicated study.

Our aim was to report a method able to measure the heterogeneity of an aggregation process in a quantitative way and in connection to the real-time observation of the growing morphology.

vi) The authors describe at the beginning of the results section two types of spherulites, viz, isotropic and anisotropic. The isotropic one is supposed to be spherical and the anisotropic ones are non-symmetric. However, looking at the images none of these looks spherical to me. The image of the small spherulite in the left side of bottom panel in Figure 1 is the closest to being a sphere. Of course, some of the images appear asymmetric than others. Hence, it would be helpful if the authors can describe the criterion being used to define the two distinct classes.

Thanks for this comment that allowed us to clarify better the nomenclature. Isotropic spherulites have been defined by us and in the literature as spherical aggregates grown isotropically and radially from a central core. The high-fidelity readouts of REPLOM revealed that under the same conditions spherulite can be initially growing linearly before it successively branches to form radially oriented amyloid fiber-like structures. These are supported by 3D-dSTORM at different time scales. The actual classification was relying on manual inspection of spherulites throughout the growing process. We note that if final structures only were recorded (see also Figure S5) this classification would be very challenging with diffraction limited microscopy. The full potential of REPLOM allows to extend beyond the final structure and records its development. The criterion has been described in the revised manuscript (line 82).

Changes in the manuscript

Added a sentence in the manuscript (line 82) to describe the nomenclature: "According to their growth pathway and the final structures, we named them isotropic spherulites and anisotropic spherulites".

vii) The authors have observed at least three different energy barriers corresponding to r_1 , r_2 and r . This is highly informative. Therefore, change of temperature should in principle alter the relative abundance of the types of spherulites observed. I would like to know if the authors have observed such differences.

This is a great comment. We have checked the relative abundance of the types of spherulites at each incubation temperature and added them in Table S2. We note that the highly heterogeneous density of particles on the field of view compounded with the low copy number of aggregates would make a quantitative assessment of their actual abundance potentially prone to artifacts. As a consequence, a tailored approach should be developed to extract this quantitative information, requiring extensive morphology mapping over a large length scale and over different fields of views combined with a more advanced automated systems for the recognition of the different species. This is out of the scope of the present work, but definitively represents a perspective for further extending the outcomes of our method. We found small variations in the percentages of isotropic and anisotropic spherulites between 45 and 37 degrees. Lowering the temperature to 32 degrees results in an increase in the abundance of isotropic spherulites from 33% to 45%. This is consistent with isotropic spherulites with lower energy barrier are more favorable to form at low incubation temperature.

Changes in the manuscript

a) Added Table S2 to show the relative abundance of the types of spherulites at each incubation temperature.

b) Added a paragraph in the manuscript (line 299) to describe the relationship between spherulite types and incubation temperature: "Table S2 shows the relative abundance of the types of spherulites at each incubation temperature. While one has to be careful in accessing the abundance of morphologies in such heterogeneous samples and the abundance of linear aggregates (fibrils and fibril-like linear core) are too low for quantitative assessment (4-9 structure in ~20 fields of view), we found small variations in the percentages of isotropic and anisotropic spherulites between 45 and 37 degree. Lowering the temperature to 32 degree results in an increase in the abundance of isotropic spherulites from 33% to 45%. This is consistent with isotropic spherulites with lower energy barrier are more favorable to form at low incubation temperature".

Bibliography

- 1 Yagi, H., Ban, T., Morigaki, K., Naiki, H. & Goto, Y. Visualization and Classification of Amyloid β Supramolecular Assemblies. *Biochemistry* **46**, 15009-15017, doi:10.1021/bi701842n (2007).
- 2 Andersen, C. B. *et al.* Branching in Amyloid Fibril Growth. *Biophysical Journal* **96**, 1529-1536, doi:<http://doi.org/10.1016/j.bpj.2008.11.024> (2009).
- 3 Huang, B., Wang, W., Bates, M. & Zhuang, X. Three-Dimensional Super-Resolution Imaging by Stochastic Optical Reconstruction Microscopy. *Science* **319**, 810-813, doi:10.1126/science.1153529 (2008).
- 4 Krebs, M. R. H. *et al.* The formation of spherulites by amyloid fibrils of bovine insulin. *Proceedings of the National Academy of Sciences of the United States of America* **101**, 14420-14424, doi:10.1073/pnas.0405933101 (2004).
- 5 Domike, K. R. & Donald, A. M. Thermal Dependence of Thermally Induced Protein Spherulite Formation and Growth: Kinetics of β -lactoglobulin and Insulin. *Biomacromolecules* **8**, 3930-3937, doi:10.1021/bm7009224 (2007).
- 6 De Luca, G. *et al.* Probing ensemble polymorphism and single aggregate structural heterogeneity in insulin amyloid self-assembly. *Journal of Colloid and Interface Science* **574**, 229-240, doi:<https://doi.org/10.1016/j.jcis.2020.03.107> (2020).
- 7 Vetri, V. & Foderà, V. The route to protein aggregate superstructures: Particulates and amyloid-like spherulites. *FEBS Letters* **589**, 2448-2463, doi:<http://doi.org/10.1016/j.febslet.2015.07.006> (2015).
- 8 Foderà, V. & Donald, A. M. Tracking the heterogeneous distribution of amyloid spherulites and their population balance with free fibrils. *The European Physical Journal E* **33**, 273-282, doi:10.1140/epje/i2010-10665-4 (2010).
- 9 Foderà, V., Zacccone, A., Lattuada, M. & Donald, A. M. Electrostatics Controls the Formation of Amyloid Superstructures in Protein Aggregation. *Physical Review Letters* **111**, 108105, doi:10.1103/PhysRevLett.111.108105 (2013).
- 10 Smith, M. I., Foderà, V., Sharp, J. S., Roberts, C. J. & Donald, A. M. Factors affecting the formation of insulin amyloid spherulites. *Colloids and Surfaces B: Biointerfaces* **89**, 216-222, doi:<http://doi.org/10.1016/j.colsurfb.2011.09.018> (2012).
- 11 Vetri, V. *et al.* Ethanol Controls the Self-Assembly and Mesoscopic Properties of Human Insulin Amyloid Spherulites. *The Journal of Physical Chemistry B* **122**, 3101-3112, doi:10.1021/acs.jpcc.8b01779 (2018).
- 12 Zhou, X. *et al.* Polysorbate 80 controls Morphology, structure and stability of human insulin Amyloid-Like spherulites. *Journal of Colloid and Interface Science* **606**, 1928-1939, doi:<https://doi.org/10.1016/j.jcis.2021.09.132> (2022).

- 13 Wägele, J., De Sio, S., Voigt, B., Balbach, J. & Ott, M. How Fluorescent Tags Modify Oligomer Size Distributions of the Alzheimer Peptide. *Biophysical Journal* **116**, 227-238, doi:<https://doi.org/10.1016/j.bpj.2018.12.010> (2019).
- 14 Graziotto, M. E. *et al.* Versatile naphthalimide tetrazines for fluorogenic bioorthogonal labelling. *RSC Chemical Biology* **2**, 1491-1498, doi:10.1039/D1CB00128K (2021).
- 15 Chen, W. *et al.* Fluorescence Self-Quenching from Reporter Dyes Informs on the Structural Properties of Amyloid Clusters Formed in Vitro and in Cells. *Nano Letters* **17**, 143-149, doi:10.1021/acs.nanolett.6b03686 (2017).
- 16 Ozawa, D. *et al.* Destruction of Amyloid Fibrils of a β 2-Microglobulin Fragment by Laser Beam Irradiation*. *Journal of Biological Chemistry* **284**, 1009-1017, doi:<https://doi.org/10.1074/jbc.M805118200> (2009).
- 17 Shen, J.-L., Tsai, M.-Y., Schafer, N. P. & Wolynes, P. G. Modeling Protein Aggregation Kinetics: The Method of Second Stochasticization. *The Journal of Physical Chemistry B* **125**, 1118-1133, doi:10.1021/acs.jpcc.0c10331 (2021).
- 18 Foderà, V. *et al.* Self-Organization Pathways and Spatial Heterogeneity in Insulin Amyloid Fibril Formation. *The Journal of Physical Chemistry B* **113**, 10830-10837, doi:10.1021/jp810972y (2009).
- 19 de Oliveira, G. A. P., Cordeiro, Y., Silva, J. L. & Vieira, T. C. R. G. in *Advances in Protein Chemistry and Structural Biology* Vol. 118 (ed Rossen Donev) 289-331 (Academic Press, 2019).

Reviewers' comments:

Reviewer #1 (Remarks to the Author):

In their revised manuscript the authors have comprehensively addressed my comments. There are still some typos that need fixing; Figure 4d is mis-rendered in my PDF copy of the manuscript.

Reviewer #2 (Remarks to the Author):

The authors have thoroughly addressed the comments raised and the manuscript is recommended for publication in Communications Biology.

Reviewer #3 (Remarks to the Author):

The authors have addressed my concerns satisfactorily. I am a bit surprised that the exposure of excitation light had no effect on the kinetics of growth of the spherulites. In Figure S14 the width of the PSF was assumed to be $\lambda/2$, I think that the authors should have used $\lambda/(2 \cdot NA)$. I am curious to know if the authors have actually measured the width of the PSF in the case of conventional microscopy. I am curious to know the resolution obtained by the authors in the superresolution microscopy. I think that it would be appropriate to mention these numbers in the paper.

Reviewer response

Reviewer #1 (Remarks to the Author):

In their revised manuscript the authors have comprehensively addressed my comments. There are still some typos that need fixing; Figure 4d is mis-rendered in my PDF copy of the manuscript.

Response

We thank the reviewer for the comment. We have fixed the typos in the revised manuscript. And we have double checked Figure 4d, it looks ok in the manuscript.

Changes in the manuscript

We have fixed a few typos and also fixed American vs British English.

Reviewer #2 (Remarks to the Author):

The authors have thoroughly addressed the comments raised and the manuscript is recommended for publication in Communications Biology.

Response

We thank the reviewer for acknowledging the impact and recommending publication to communication biology.

Reviewer #3 (Remarks to the Author):

The authors have addressed my concerns satisfactorily. I am a bit surprised that the exposure of excitation light had no effect on the kinetics of growth of the spherulites.

Response

We thank the reviewer for the comment. This is indeed a good point that we had rectified by conditioning our experimental setup. We kept the ratio of labeled to unlabeled insulin very low (1/10000). This combined with the low frame rate of 30ms exposure every 0.5 min minimized photodamage. Please note that this results in a few hundred frames for a several hour experiment. As such the exposure to light is significantly lower than classical single particle experiments or imaging in cells. Considering the above reasons, we think it's normal that the excitation light had no effect on the kinetics of growth of spherulites. We further discussed this in the revised manuscript.

Changes in the manuscript

We discussed the reason in the revised manuscript (line 203): "This could be due to the low labeled to unlabeled insulin ratio (1/10000) and the very limited exposure to excitation light (30ms every 20-40 s) resulting in a few hundred frames per experiment."

In Figure S14 the width of the PSF was assumed to be $\lambda/2$, I think that the authors should have used $\lambda/(2*NA)$.

Response

Thank you for noticing the coding error. We have rectified this now and replotted the figure, which has minuscule difference from the previous one. The conclusion is practically identical.

Changes in the manuscript

We have replaced Figure S14 with the new plotted figure.

I am curious to know if the authors have actually measured the width of the PSF in the case of conventional microscopy.

Response

Thanks for the comment. We have measured the PSF of conventional microscopy now and have described it in the method section (line 377).

Changes in the manuscript

We have add a paragraph to describe the PSF of conventional microscopy in line 377: "The PSF of our TIRF microscopy was determined by the FWHM of Alexa Fluor 647 labeled insulin imaged under otherwise identical condition. In detail, Alexa Fluor 647 labeled monomeric insulin was first filtered through a 0.22 μm filter, then added on the poly-L-lysine surface at a low density to ensure a minimal signal overlap. The average FWHM was measured to be about 383 nm, slightly (30%) higher than the calculated under optimal conditions.

I am curious to know the resolution obtained by the authors in the superresolution microscopy. I think that it would be appropriate to mention these numbers in the paper.

Response

Thanks for the great comment. Actually, we have calculated the resolution of REPLOM by the FWHM of single spots in the reconstructed image, see Methods "Resolution of REPLOM" (line 440) and Figure S10. Acknowledging this may not be clear we have added an extra sentence highlighting the resolution in the main text.

Changes in the manuscript

We have a sentence in line 191: "The resolution of REPLOM was determined by the FWHM of multiple spots' intensity, which was ~ 66 nm (see Methods and Figure S10)."

REVIEWERS' COMMENTS:

Reviewer #3 (Remarks to the Author):

The authors have addressed all the queries satisfactorily.